# An Information-theoretical Framework for Understanding Out-of-distribution Detection with Pretrained Vision-Language Models

**Bo Peng, Jie Lu, Guangquan Zhang, Zhen Fang**[*]
University of Technology Sydney

## Abstract

Out-of-distribution (OOD) detection, recognized for its ability to identify samples of unknown classes, provides solid advantages in ensuring the reliability of machine learning models. Among existing OOD detection methods, pre-trained vision-language models have emerged as powerful post-hoc OOD detectors by leveraging textual and visual information. Despite the empirical success, there still remains a lack of research on a formal understanding of their effectiveness. This paper bridges the gap by theoretically demonstrating that existing CLIP-based post-hoc methods effectively perform a stochastic estimation of the point-wise mutual information (PMI) between the input image and each in-distribution label. This estimation is then utilized to construct energy functions for modeling in-distribution distributions. Different from prior methods that inherently consider PMI estimation as a whole task, we, motivated by the divide-and-conquer philosophy, decompose PMI estimation into multiple easier sub-tasks by applying the chain rule of PMI, which not only reduces the estimation complexity but also provably increases the estimation upper bound to reduce the underestimation bias. Extensive evaluations across mainstream benchmarks empirically manifest that our method establishes a new state-of-the-art in a variety of OOD detection setups.

## 1 Introduction

Despite the significant progress in machine learning that has facilitated a broad spectrum of classification tasks [2, 79, 41], models often operate under a *closed-world* scenario, where test data stems from the same distribution as the training data. However, real-world applications often entail *open-world* scenarios in which deployed models may encounter unseen classes of samples during training, giving rise to what is known as out-of-distribution (OOD) data. These OOD instances can potentially undermine a model's stability and, in certain cases, inflict severe damage on its performance. Accordingly, a reliable discriminative model should not only correctly classify known in-distribution (ID) samples but also flag any OOD inputs as unknown. This directly motivates OOD detection [58, 72, 28] which ensures the safety of decision-critical applications [25, 87].

This paper focuses on post-hoc OOD detection, which are more practical than learning-based methods that require resource-intensive retraining. Earlier studies [20, 34, 61, 36] primarily utilized the single modality of pre-trained models, but the success of contrastive language-image pre-training (CLIP) [57] has recently shifted research toward expanding post-hoc OOD detection from single-modal to multi-modal methods. Researchers have since explored ways to better leverage multi-modal models to enhance the performance and applicability of post-hoc OOD detection. A notable method is MCM [42], which defines textual features as concept prototypes for each ID class and uses the scaled distance between visual features and the closest ID prototype to measure OOD uncertainty. This

---

[*]Correspondence to Zhen Fang (zhen.fang@uts.edu.au)

39th Conference on Neural Information Processing Systems (NeurIPS 2025).

method has paved the way for using pre-trained vision-language models (VLMs) in post-hoc OOD detection. However, MCM relies only on textual information from the ID label space, leaving VLMs' text interpretation capabilities underutilized. To address this, NegLabel [26] introduces numerous negative labels, allowing the model to better distinguish OOD samples. A heuristic grouping strategy in NegLabel is also proposed to further enhance OOD detection performance. Despite its promising potential, it is worth noting that a formalized understanding of CLIP-based post-hoc OOD detection remains significantly lacking in the field. This prompts the research question underlying this work:

> *How to theoretically justify the empirical effectiveness of CLIP-based post-hoc OOD detection?*

**Theoretical Significance.** To address this challenge, we draw inspiration from information theory and propose an information-theoretical density-based framework. In this framework, ID data is modeled as an energy-based model, where the point-wise mutual information (PMI) [6] between the input image and each ID label forms the energy functions. We argue that this analytical framework is well-suited for studying OOD detection, as OOD data, by definition, inherently diverges from ID data in terms of their underlying density distributions. Guided by this framework, we show that representative CLIP-based post-hoc OOD detection methods [42, 26] can be interpreted as stochastic Monte Carlo estimations of PMI. Furthermore, we theoretically establish the following key points: 1) introducing negative labels increases the estimation upper bound, thereby mitigating underestimation bias; and 2) the grouping strategy effectively approximates the expectation through multiple sampling, reducing estimation variance.

**Algorithmic Contribution.** To further facilitate PMI estimation for OOD scoring, the starting point of our method is to decompose PMI as a sum of terms by applying the chain rule on PMI. In addition to reduce the overall estimation complexity according to the *divide-and-conquer* philosophy, we prove that the decomposed PMI estimation can further increases the estimation upper bound to reduce underestimation bias without explicitly introduce a corresponding number of negative labels. Notably, NegLabel [26] has empirically found that introducing excessive negative label would degrade OOD detection performance.

## 2    Related work

The core of CLIP-based OOD detection lies in how to leverage texture supervision with pre-trained VLMs to assist OOD detection on the visual domain. On the one hand, the pioneering work, MCM [42], defines textual features as concept proto- types for each ID class and uses the scaled distance between visual features and the closest ID prototype to measure OOD uncertainty. Intead of relying only on textual information from the ID label space, NegLabel [26] incorporates additional negative class names mined from available data sources, such as WordNet, as negative proxies. To mitigate the nonalignment between target visual OOD distribution and the generated negative textual OOD distribution, AdaNeg [75] leverages the benefits of test-time adaptation to generate adaptive proxies by exploring potential OOD images during testing. On the other hand, CLIP-based OOD detection can also be improved by prompt representation learning. In particular, LoCoOp [43] learns ID text prompts by pushing them away from the portions of CLIP local features that have ID-irrelevant nuisances (e.g., backgrounds). CLIPN [68] and LSN [46] design a learnable "no" prompt and a "no" text encoder to capture negation semantics within images. Differently, LAPT [76] initializes prompts with negative labels [26], followed by tuning prompts with cross-modal and cross-distribution mixing. *Due to limited space, related works on traditional OOD detection are discussed in Appendix A.*

## 3    Preliminary

**Notations.** Let $\mathcal{X}$ and $\mathcal{Y}$ represent the input space and the label space, respectively. Considering two random variables $X \in \mathcal{X}$ and $Y \in \mathcal{Y}$, we represent $p_{XY}$ and $p_X$ as the probability *density* functions of the joint distribution $\mathbb{P}_{XY}$ and the marginal distribution $\mathbb{P}_X$, respectively. Similarly, $p_Y$ and $p_{Y|X}$ denote the *mass* functions of marginal and conditional distributions $\mathbb{P}_Y$ and $\mathbb{P}_{Y|X}$, respectively. We write $\mathbb{P}_{X_I Y_I}$ as the joint ID distribution defined over $\mathcal{X} \times \mathcal{Y}_I$, where $\mathcal{Y}_I \triangleq \{y_1, \ldots, y_K\} \subset \mathcal{Y}$ is the

space for *known* ID labels. During testing, there are some unknown OOD joint distributions $\mathbb{P}_{X_o Y_o}$ defined over $\mathcal{X} \times \mathcal{Y}_o$, where $\mathcal{Y}_o \triangleq \mathcal{Y} \setminus \mathcal{Y}_I$ presents the space of *unknown* OOD labels.

**Post-hoc Detection Strategy.** Concurrently, OOD detection follows a training-free scoring mechanism, *i.e.,* given a pre-trained ID classification model parameterized by $\boldsymbol{\theta}$, and a scoring function $S$, then $\mathbf{x}$ is detected as ID data if and only if $S(\mathbf{x}; \boldsymbol{\theta}) \geq \lambda$, for some given threshold $\lambda$:

$$g(\mathbf{x}) = \text{ID, if } S(\mathbf{x}; \boldsymbol{\theta}) \geq \lambda; \text{ otherwise, } g(\mathbf{x}) = \text{OOD}, \tag{1}$$

where $\lambda$ is chosen to correctly classify a high fraction of ID data (e.g., 95%).

**CLIP-based Models.** Given any visual input $\mathbf{x} \in \mathcal{X}$ and any label $y \in \mathcal{Y}$, we extract features of $\mathbf{x}$ and $y$ using an arbitrary CLIP-based model $\mathbf{f}$ that consists an image encoder $\mathbf{f}(\cdot; \boldsymbol{\theta}_{\text{img}})$ and an text encoder $\mathbf{f}(\cdot; \boldsymbol{\theta}_{\text{text}})$ as follows:

$$\mathbf{z} = \mathbf{f}(\mathbf{x}; \boldsymbol{\theta}_{\text{img}}) \in \mathbb{S}^{d-1}, \quad \mathbf{r} = \mathbf{f}(\mathcal{Q}(y); \boldsymbol{\theta}_{\text{text}}) \in \mathbb{S}^{d-1},$$

where $\mathcal{Q}(\cdot)$ is the text prompt template, $\mathbb{S}^{d-1} \triangleq \{\mathbf{r} \in \mathbb{R}^d | \|\mathbf{r}\|_2 = 1\}$, and $\boldsymbol{\theta} = \{\boldsymbol{\theta}_{\text{img}}, \boldsymbol{\theta}_{\text{text}}\}$.

**CLIP-based OOD Detectors Studied.** CLIP-based models, which are initially proposed for zero-shot ID classification, have recently been extended to zero-shot OOD detection where there is no need to train on ID samples. The pioneering work, MCM [42], treats the prompt of ID labels as concept prototypes and measures the ID-ness of the input image by comparing the similarity between the input image and the concept prototypes in the feature space learned by CLIP-based models, i.e.,

$$S_{\text{MCM}}(\mathbf{x}; \boldsymbol{\theta}) \triangleq \max_{y \in \mathcal{Y}_I} \frac{\exp(\mathbf{z}^\top \mathbf{r}/\tau)}{\sum_{y \in \mathcal{Y}_I} \exp(\mathbf{z}^\top \mathbf{r}_j/\tau)}, \tag{2}$$

where $\tau > 0$ is a temperature hyper-parameter. Unlike MCM that only employs information from the ID label space, NegLabel [26] introduces a $L$-sized set of negative labels[2] $\{y_{K+1}, \ldots, y_{K+L}\}$ sourced from lexical databases, followed by randomly grouping the selected $L$ negative labels into $T$ *non-overlapping* subsets $\mathcal{G}_1, ..., \mathcal{G}_T$, i.e., $\mathcal{G}_i \cap \mathcal{G}_j = \emptyset, \forall i \neq j$:

$$S_{\text{NegLabel}}(\mathbf{x}; \boldsymbol{\theta}) \triangleq \frac{1}{T} \sum_{t=1}^{T} \sum_{y \in \mathcal{Y}_I} \frac{\exp(\mathbf{z}^\top \mathbf{r}/\tau)}{\sum_{y_j \in \mathcal{G}_t \cup \mathcal{Y}_I} \exp(\mathbf{z}^\top \mathbf{r}_j/\tau)}. \tag{3}$$

## 4  Theoretical Analysis

While both MCM and NegLabel have empirically emerged to be effective post-hoc OOD detectors, their inherent connections and theoretical understandings are largely lacking. To the best of our knowledge, there is limited prior work providing provable guarantees for CLIP-based post-hoc OOD detection methods from a rigorous mathematical point of view. In this section, we provide theoretical justification for CLIP-based post-hoc OOD detection from the perspective of *information-theoretical density estimation*. In particular, due to the fact that OOD data, by definition, inherently diverges from ID data by means of their data density distributions, we, following advanced density-based OOD detection methods [51, 44, 36], render the ID density function as an ideal metric for ID-OOD discrimination. Inspired by prior works [36], we consider modeling the unknown true ID density function $p_{X_I}$ of ID input marginal distribution $\mathbb{P}_{X_I}$ by resorting to the energy-based model [29, 19]:

$$\hat{p}_{X_I}(\mathbf{x}) = \frac{\exp[E(\mathbf{x})]}{Z} \propto \exp[E(\mathbf{x})], \quad E(\mathbf{x}) = \frac{1}{\alpha} \log \sum_{y \in \mathcal{Y}_I} \exp[\alpha \mathcal{W}(\mathbf{x}, y)], \tag{4}$$

where $Z = \int \exp[E(\mathbf{x})] \, d\mathbf{x}$ is an *input-independent* normalization function, $\alpha > 0$ is a hyper-parameter and $\mathcal{W}(\mathbf{x}, y)$ is the *point-wise mutual information* (PMI) [6] that explicitly measures the association between the input $\mathbf{x} \in \mathcal{X}$ and the label $y \in \mathcal{Y}$. As implied by the following definition of PMI, the formulation in Eq. (4) implicitly assumes that the modelled ID density $\hat{p}_{X_I}(\mathbf{x})$ is induced by an underlying unknown joint distribution $\mathbb{P}_{XY}$ defined over $\mathcal{X} \times \mathcal{Y}$.

---

[2]In accordance to Jiang et al. [26], labels $y \in \mathcal{Y}_o$ that have lower affinities with ID images compared to OOD images are considered as negative labels

**Definition 1.** The PMI between two observations $\mathbf{x} \in \mathcal{X}$ and $y \in \mathcal{Y}$ is defined as follows:

$$\mathcal{W}(\mathbf{x}, y) \triangleq \log \frac{p_{XY}(\mathbf{x}, y)}{p_X(\mathbf{x}) p_Y(y)} = \log \frac{p_{Y|X}(y|\mathbf{x})}{p_Y(y)}. \tag{5}$$

Note that directly calculating $\mathcal{W}(\mathbf{x}, y)$ in Eq. (5), which is built upon the conditional distribution $\mathbb{P}_{Y|X}$ and the marginal distribution $\mathbb{P}_Y$, can be computationally intractable since the two underlying distributions are unknown in nearly all practical applications. In response to this challenge, our key idea is to replace the unknown mass function $p_{Y|X}(y|\mathbf{x})$ with the estimated one $\hat{p}_{Y|X}(y|\mathbf{x}; \boldsymbol{\theta})$ using the pre-trained CLIP-based model parameters $\boldsymbol{\theta}$ for a tractable estimator of the modeled ID data density function $\hat{p}_{X_{\mathrm{I}}}(\mathbf{x})$ in Eq. (4), i.e.,

$$\hat{p}_{X_{\mathrm{I}}}(\mathbf{x}; \boldsymbol{\theta}) = \frac{\exp\left[E_{\boldsymbol{\theta}}(\mathbf{x})\right]}{Z_{\boldsymbol{\theta}}} \propto \exp\left[E_{\boldsymbol{\theta}}(\mathbf{x})\right], \quad E_{\boldsymbol{\theta}}(\mathbf{x}) = \frac{1}{\alpha} \log \sum_{y \in \mathcal{Y}_{\mathrm{I}}} \exp\left[\alpha \hat{\mathcal{W}}(\mathbf{x}, y)\right], \tag{6}$$

where $Z_{\boldsymbol{\theta}} = \int \exp\left[E_{\boldsymbol{\theta}}(\mathbf{x})\right] d\mathbf{x}$ and $\hat{\mathcal{W}}(\mathbf{x}, y; \boldsymbol{\theta}) = \log \frac{\hat{p}_{Y|X}(y|\mathbf{x}; \boldsymbol{\theta})}{p_Y(y)}$ is the estimator[3] of the true PMI $\mathcal{W}(\mathbf{x}, y)$. In the following, we demonstrate that MCM and NegLabel, despite their seemingly distinct scoring functions, can be interpreted as methods for stochastically estimating PMI and therefore the energy function that effectively replicates the behavior of the ID density.

## 4.1 Towards Understanding MCM

To tractably derive $\hat{p}_{Y|X}(y|\mathbf{x}; \boldsymbol{\theta})$ in Eq. (6), we, motivated by prior works [56, 11], assume that $\hat{p}_{Y|X}(y|\mathbf{x}; \boldsymbol{\theta})$ belongs to a energy-based variational family that uses a critic $h_{\boldsymbol{\theta}}$ parameterized by $\boldsymbol{\theta}$ and is scaled by the marginal mass function $p_Y$, i.e.,

$$\hat{p}_{Y|X}(y|\mathbf{x}; \boldsymbol{\theta}) = \frac{p_Y(y) \exp h_{\boldsymbol{\theta}}(\mathbf{x}, y)}{\sum_{\hat{y} \in \mathcal{Y}} p_Y(\hat{y}) \exp h_{\boldsymbol{\theta}}(\mathbf{x}, \hat{y})} = \frac{p_Y(y) \exp h_{\boldsymbol{\theta}}(\mathbf{x}, y)}{\Phi_{\boldsymbol{\theta}}(\mathbf{x})}, \tag{7}$$

where $\Phi_{\boldsymbol{\theta}}(\mathbf{x}) = \mathbb{E}_{\hat{y} \sim \mathbb{P}_Y}\left[\exp h_{\boldsymbol{\theta}}(\mathbf{x}, \hat{y})\right]$ is the normalization function. Following Peng et al. [51], one can consider a Monte-Carlo method to construct a simple and analytically tractable estimator of $\Phi_{\boldsymbol{\theta}}(\mathbf{x})$ by sampling a $N$-sized set of *i.i.d.* samples $\hat{\mathbf{y}}_N = \{\hat{y}_1, ..., \hat{y}_N\} \sim \mathbb{P}_Y^N$, i.e.,

$$\Phi_{\boldsymbol{\theta}}(\mathbf{x}) \approx \frac{1}{N} \sum_{j=1}^{N} \exp h_{\boldsymbol{\theta}}(\mathbf{x}, \hat{y}_j), \tag{8}$$

which implies that

$$\log \frac{\hat{p}_{Y|X}(y|\mathbf{x}; \boldsymbol{\theta})}{p_Y(y)} \approx \log \frac{N \exp h_{\boldsymbol{\theta}}(\mathbf{x}, y)}{\sum_{j=1}^{N} \exp h_{\boldsymbol{\theta}}(\mathbf{x}, \hat{y}_j)}. \tag{9}$$

If we set $h_{\boldsymbol{\theta}}(\mathbf{x}, y) = \mathbf{z}^\top \mathbf{r}/\tau$ and $\hat{\mathbf{y}}_N = \mathcal{Y}_{\mathrm{I}}$ such that $N = K$ in Eq. (9), in the extreme case where $\alpha \to +\infty$, combining Eq. (9) and Eq. (6) implies that

$$
\begin{aligned}
\lim_{\alpha \to +\infty} E_{\boldsymbol{\theta}}(\mathbf{x}) &= \lim_{\alpha \to +\infty} \frac{1}{\alpha} \log \sum_{y \in \mathcal{Y}_{\mathrm{I}}} \exp\left[\alpha \log \frac{\hat{p}_{Y|X}(y|\mathbf{x}; \boldsymbol{\theta})}{p_Y(y)}\right] \\
&= \max_{y \in \mathcal{Y}_{\mathrm{I}}} \log \frac{\hat{p}_{Y|X}(y|\mathbf{x}; \boldsymbol{\theta})}{p_Y(y)} \\
&\approx \max_{y \in \mathcal{Y}_{\mathrm{I}}} \log \frac{K \cdot \exp h_{\boldsymbol{\theta}}(\mathbf{x}, y)}{\sum_{j=1}^{K} \exp h_{\boldsymbol{\theta}}(\mathbf{x}, y_j)} \\
&= \log \underbrace{\max_{y \in \mathcal{Y}_{\mathrm{I}}} \frac{\exp(\mathbf{z}^\top \mathbf{r}/\tau)}{\sum_{j=1}^{K} \exp(\mathbf{z}^\top \mathbf{r}_j/\tau)}}_{S_{\mathrm{MCM}}(\mathbf{x}; \boldsymbol{\theta})} + \underbrace{\log K}_{\text{const}}.
\end{aligned} \tag{10}
$$

Since the logarithm function is monotonically increasing, Eq. (10) implies that $S_{\mathrm{MCM}}(\mathbf{x}; \boldsymbol{\theta})$ in Eq. (2) can be understood as a stochastic estimator of $\lim_{\alpha \to +\infty} E_{\boldsymbol{\theta}}(\mathbf{x})$ (up to a constant). Theorem 1 provides provable guarantees of how $S_{\mathrm{MCM}}(\mathbf{x}; \boldsymbol{\theta})$ correctly recovers the true energy function $E(\mathbf{x})$ in Eq. (4) when $\alpha \to +\infty$.

---

[3] As we will demonstrate later, the mass function $p_Y$ can cancel out during the calculation of $\hat{\mathcal{W}}(\mathbf{x}, y; \boldsymbol{\theta})$ so that there is no need to estimate $p_Y$.

**Theorem 1.** *Let $h_{\boldsymbol{\theta}}(\mathbf{x}, y) = \mathbf{z}^\top \mathbf{r}/\tau$ and $N = K$. If we, following prior works [56, 47], assume that $h_{\boldsymbol{\theta}}(\mathbf{x}, y) = \log \frac{p_{Y|X}(y|\mathbf{x})}{p_Y(y)} + c(\mathbf{x})$ with $c(\mathbf{x})$ as a constant term depending on $\mathbf{x}$, in the extreme case where $\alpha \to +\infty$, then we have the following[4]:*

$$\lim_{\alpha \to +\infty} E(\mathbf{x}) = \lim_{\alpha \to +\infty} E_{\boldsymbol{\theta}}(\mathbf{x}) = \lim_{N \to +\infty} \log N S_{MCM}(\mathbf{x}; \boldsymbol{\theta}). \tag{11}$$

## 4.2 Towards Understanding NegLabel

To study NegLabel theoretically, we make the ID data density estimation depend on multiple samples. In particular, given a set of *i.i.d.* samples $\hat{\mathbf{y}}_N = \{\hat{y}_1, ..., \hat{y}_N\} \sim \mathbb{P}_Y^N$, we can rewrite $\hat{p}_{Y|X}(y|\mathbf{x}; \boldsymbol{\theta})$ as:

$$\hat{p}_{Y|X}(y|\mathbf{x}; \boldsymbol{\theta}) = \mathbb{E}_{\hat{\mathbf{y}}_N \sim \mathbb{P}_Y^N} \left[ \hat{p}_{Y|X\mathbf{Y}_N}(y|\mathbf{x}, \hat{\mathbf{y}}_N; \boldsymbol{\theta}) \right], \tag{12}$$

where $\mathbf{Y}_N = (Y_1, Y_2, \ldots, Y_N)$ is an $N$-dimensional random variable with each $Y_i$ as an i.i.d. copy of the random variable $Y$. Motivated by Poole et al. [56], we then model the term $\hat{p}_{Y|X\mathbf{Y}_N}(y|\mathbf{x}, \hat{\mathbf{y}}_N; \boldsymbol{\theta})$ in Eq. (12) as follows[5]:

$$\hat{p}_{Y|X\mathbf{Y}_N}(y|\mathbf{x}, \hat{\mathbf{y}}_N; \boldsymbol{\theta}) = \frac{p_Y(y) \exp h_{\boldsymbol{\theta}}(\mathbf{x}, y)}{\Psi_{\boldsymbol{\theta}}(\mathbf{x}, y, \hat{\mathbf{y}}_N)/(N+1)}, \tag{13}$$

where

$$\Psi_{\boldsymbol{\theta}}(\mathbf{x}, y, \hat{\mathbf{y}}_N) = \exp h_{\boldsymbol{\theta}}(\mathbf{x}, y) + \sum_{j=1}^{N} \exp h_{\boldsymbol{\theta}}(\mathbf{x}, \hat{y}_j).$$

If we set $h_{\boldsymbol{\theta}}(\mathbf{x}, y) = \mathbf{z}^\top \mathbf{r}/\tau$, $N = K + L/T$ and $\alpha = 1$, combining Eq. (12) and Eq. (13) with Eq. (6) directly results in the following:

$$
\begin{aligned}
E_{\boldsymbol{\theta}}(\mathbf{x}) &= \log \sum_{y \in \mathcal{Y}_{\mathrm{I}}} \frac{\hat{p}_{Y|X}(y|\mathbf{x}; \boldsymbol{\theta})}{p_Y(y)} \\
&= \log \sum_{y \in \mathcal{Y}_{\mathrm{I}}} \mathbb{E}_{\hat{\mathbf{y}}_{N-1} \sim \mathbb{P}_Y^{N-1}} \left[ \frac{\exp h_{\boldsymbol{\theta}}(\mathbf{x}, y)}{\Psi_{\boldsymbol{\theta}}(\mathbf{x}, y, \hat{\mathbf{y}}_{N-1})} \right] + \log N \\
&\approx \log \underbrace{\sum_{y \in \mathcal{Y}_{\mathrm{I}}} \frac{1}{T} \sum_{t=1}^{T} \frac{\exp(\mathbf{z}^\top \mathbf{r}/\tau)}{\sum_{y_j \in \mathcal{G}_t \cup \mathcal{Y}_{\mathrm{I}}} \exp(\mathbf{z}^\top \mathbf{r}_j/\tau)}}_{S_{\mathrm{NegLabel}}(\mathbf{x};\boldsymbol{\theta})} + \underbrace{\log(K + L/T)}_{\mathrm{const}}.
\end{aligned}
\tag{14}
$$

Since the logarithm function is monotonically increasing, Eq. (14) implies that $S_{\mathrm{NegLabel}}(\mathbf{x}; \boldsymbol{\theta})$ can be interpreted as another Monte-Carlo estimator of $E_{\boldsymbol{\theta}}(\mathbf{x})$ (up to a constant) by sampling $\hat{\mathbf{y}}_{N-1}$ from $\mathbb{P}_Y^{N-1}$ $T$ times with $N = K + L/T$, where, for each $y \in \mathcal{Y}_{\mathrm{I}}$, samples from $\mathcal{G}_t \cup \mathcal{Y}_{\mathrm{I}} \setminus \{y\}$ are instantiated as $\hat{\mathbf{y}}_{N-1}$ on the $t$-th round of sampling $(1 \le t \le T)$[6]. Theorem 2 provides provable guarantees of how $S_{\mathrm{NegLabel}}(\mathbf{x}; \boldsymbol{\theta})$ correctly recover $E(\mathbf{x})$ in Eq. (4).

**Theorem 2.** *Let $h_{\boldsymbol{\theta}}(\mathbf{x}, y) = \mathbf{z}^\top \mathbf{r}/\tau$, $\alpha = 1$, and $N = K + L/T$. If we, following following prior works [56, 47], assume that $h_{\boldsymbol{\theta}}(\mathbf{x}, y) = \log \frac{p_{Y|X}(y|\mathbf{x})}{p_Y(y)} + c(\mathbf{x})$ with $c(\mathbf{x})$ as a constant term depending on $\mathbf{x}$, then we have the following[7]:*

$$E(\mathbf{x}) = \lim_{N \to +\infty} E_{\boldsymbol{\theta}}(\mathbf{x}) = \lim_{\substack{T \to +\infty \\ L/T \to +\infty}} \log(K + L/T) S_{NegLabel}(\mathbf{x}; \boldsymbol{\theta}). \tag{15}$$

**Remark.** To reveal how negative labels benefit OOD detection, let us looking to Eq.(14) where the estimated energy function $E_{\boldsymbol{\theta}}(\mathbf{x})$ is upper bounded by $\log N$ with $N$ as the number of labels drawn from $\mathbb{P}_Y$. By introducing negative labels to take $N = K + L/T$, the estimator $E_{\boldsymbol{\theta}}(\mathbf{x})$ are allowed to capture at most $\log(K + L/T)$ nats of $E(\mathbf{x})$, which is strictly larger than $\log K$ of $S_{\mathrm{MCM}}(\mathbf{x}; \boldsymbol{\theta})$ in Eq. (9). On the other hand, Theorem 2 states that the recovery of $E(\mathbf{x})$ requires $N = K + L/T \to +\infty$, which theoretically justified the use of negative labels in OOD scoring.

---

[4]We detail the derivation in Appendix B

[5]We justify this formulation in Appendix C

[6]More details can be found in Step 3 of Appendix D

[7]We detail the derivation in Appendix D.

# 5 Methodology

Based on the thorectical analysis in Section 4, one may conclude that MCM and NegLabel can be regarded to formulate the estimation of PMI as a whole task. Differently, inspired by the *divide-and-conquer* philosophy, we conjecture that PMI estimation could be simplified as well as improved by decoupling the task into multiple earlier sub-tasks. Central to our method, we introduce an auxiliary random variable $\tilde{X} = \mathcal{T}(X) \in \mathcal{X}$, whose realization is denoted by $\tilde{\mathbf{x}}$, to represent sub-views[8] of the random variable $X \in \mathcal{X}$ with $\mathcal{T}$ as a transformation function. To be specific, given the input $\mathbf{x}$ is an image, we can create a sub-view $\tilde{\mathbf{x}}$ by randomly either 1) occluding some of the pixels in $\mathbf{x}$ with $\mathcal{T}$ as *Cutout* [10] or 2) cropping a random portion of $\mathbf{x}$ with $\mathcal{T}$ as *Random Cropping*. In the rest of this paper, we consider the latter case as the default setting.

**Theorem 3.** *For any $\mathbf{x} \in \mathcal{X}$ and $y \in \mathcal{Y}$, let $\tilde{\mathbf{x}} = \mathcal{T}(\mathbf{x})$ be a sub-view of the input $\mathbf{x}$, $\mathcal{W}(\mathbf{x}, y)$ can be decomposed into the following two terms[9]:*

$$\mathcal{W}(\mathbf{x}, y) = \mathcal{W}(\tilde{\mathbf{x}}, y) + \mathcal{W}(\mathbf{x}, y|\tilde{\mathbf{x}}), \tag{16}$$

*where $\mathcal{W}(\mathbf{x}, y|\tilde{\mathbf{x}})$, i.e., the PMI between $\mathbf{x}$ and $y$ conditioned on $\tilde{\mathbf{x}}$, is defined as follows:*

$$\begin{aligned}
\mathcal{W}(\mathbf{x}, y|\tilde{\mathbf{x}}) &\triangleq \log \frac{p_{XY|\tilde{X}}(\mathbf{x}, y|\tilde{\mathbf{x}})}{p_{X|\tilde{X}}(\mathbf{x}|\tilde{\mathbf{x}})p_{Y|\tilde{X}}(y|\tilde{\mathbf{x}})} \\
&= \log \frac{p_{Y|X\tilde{X}}(y|\mathbf{x}, \tilde{\mathbf{x}})}{p_{Y|\tilde{X}}(y|\tilde{\mathbf{x}})}.
\end{aligned} \tag{17}$$

Let $\hat{\mathcal{W}}(\tilde{\mathbf{x}}, y; \boldsymbol{\theta})$ and $\hat{\mathcal{W}}(\mathbf{x}, y|\tilde{\mathbf{x}}; \boldsymbol{\theta})$ be the estimator of $\mathcal{W}(\tilde{\mathbf{x}}, y)$ and $\mathcal{W}(\mathbf{x}, y|\tilde{\mathbf{x}})$ with the pre-trained parameters $\boldsymbol{\theta}$, respectively, Theorem 3 directly implies that we can rewrite $\hat{\mathcal{W}}(\mathbf{x}, y; \boldsymbol{\theta})$ as follows:

$$\hat{\mathcal{W}}(\mathbf{x}, y; \boldsymbol{\theta}) = \hat{\mathcal{W}}(\tilde{\mathbf{x}}, y; \boldsymbol{\theta}) + \hat{\mathcal{W}}(\mathbf{x}, y|\tilde{\mathbf{x}}; \boldsymbol{\theta}), \tag{18}$$

**Parameterizing $\hat{\mathcal{W}}(\tilde{\mathbf{x}}, y; \boldsymbol{\theta})$.** Let $\hat{p}_{Y|\tilde{X}}(y|\tilde{\mathbf{x}}; \boldsymbol{\theta})$ denote the estimator of $p_{Y|\tilde{X}}(y|\tilde{\mathbf{x}})$, according to Eq. (5), we can parameterize $\hat{\mathcal{W}}(\tilde{\mathbf{x}}, y; \boldsymbol{\theta})$ as

$$\hat{\mathcal{W}}(\tilde{\mathbf{x}}, y; \boldsymbol{\theta}) = \log \frac{\hat{p}_{Y|\tilde{X}}(y|\tilde{\mathbf{x}}; \boldsymbol{\theta})}{p_Y(y)}, \tag{19}$$

Given $\hat{\mathbf{y}}_{N-1} = \{\hat{y}_1, ..., \hat{y}_{N-1}\} \sim \mathbb{P}_Y^{N-1}$, following Eq. (12) and Eq. (13), $\hat{p}_{Y|\tilde{X}}(y|\tilde{\mathbf{x}}; \boldsymbol{\theta})$ takes the following form:

$$\hat{p}_{Y|\tilde{X}}(y|\tilde{\mathbf{x}}; \boldsymbol{\theta}) = \mathbb{E}_{\hat{\mathbf{y}}_{N-1} \sim \mathbb{P}_Y^{N-1}} \left[ \frac{p_Y(y) \exp h_{\boldsymbol{\theta}}(\tilde{\mathbf{x}}, y)}{\Psi_{\boldsymbol{\theta}}(\tilde{\mathbf{x}}, y, \hat{\mathbf{y}}_{N-1})/N} \right]. \tag{20}$$

Similar to Eq. (14), let $N = K + L/T$ and $h_{\boldsymbol{\theta}}(\tilde{\mathbf{x}}, y) = \tilde{\mathbf{z}}^\top \mathbf{r}/\tau$ with $\tilde{\mathbf{z}} = \mathbf{f}(\tilde{\mathbf{x}}; \boldsymbol{\theta}_{\text{img}})$, we can arrive at the following Monte-Carlo estimator of $\hat{\mathcal{W}}(\tilde{\mathbf{x}}, y; \boldsymbol{\theta})$ given by

$$\begin{aligned}
\hat{\mathcal{W}}(\tilde{\mathbf{x}}, y; \boldsymbol{\theta}) &= \log \mathbb{E}_{\hat{\mathbf{y}}_{N-1} \sim \mathbb{P}_Y^{N-1}} \left[ \frac{N \cdot \exp h_{\boldsymbol{\theta}}(\tilde{\mathbf{x}}, y)}{\Psi(\tilde{\mathbf{x}}, y, \hat{\mathbf{y}}_{N-1})} \right] \\
&\approx \Lambda(\tilde{\mathbf{x}}, y; \boldsymbol{\theta}) \\
&\triangleq \log \sum_{y \in \mathcal{Y}_{\text{I}}} \frac{1}{T} \sum_{t=1}^{T} \frac{\exp(\tilde{\mathbf{z}}^\top \mathbf{r}/\tau)}{\sum_{y_j \in \mathcal{G}_t \cup \mathcal{Y}_{\text{I}}} \exp(\tilde{\mathbf{z}}^\top \mathbf{r}_j/\tau)} + \log(K + L/T).
\end{aligned} \tag{21}$$

**Parameterizing $\hat{\mathcal{W}}(\mathbf{x}, y|\tilde{\mathbf{x}}; \boldsymbol{\theta})$.** Let $\hat{p}_{Y|X\tilde{X}}(y|\mathbf{x}, \tilde{\mathbf{x}}; \boldsymbol{\theta})$ be the estimator of $p_{Y|X\tilde{X}}(y|\mathbf{x}, \tilde{\mathbf{x}})$, according to Eq. (47), we can parameterize $\hat{\mathcal{W}}(\mathbf{x}, y|\tilde{\mathbf{x}}; \boldsymbol{\theta})$ as:

$$\hat{\mathcal{W}}(\mathbf{x}, y|\tilde{\mathbf{x}}; \boldsymbol{\theta}) = \log \frac{\hat{p}_{Y|X\tilde{X}}(y|\mathbf{x}, \tilde{\mathbf{x}}; \boldsymbol{\theta})}{\hat{p}_{Y|\tilde{X}}(y|\tilde{\mathbf{x}}; \boldsymbol{\theta})}. \tag{22}$$

---

[8]Sub-views are those derived from the original view without introducing any external information
[9]We detail the derivation in Appendix E

Under the assumption of a similar energy-based variational family to Eq. (7), we can formulate $\hat{p}_{Y|X\tilde{X}}(y|\mathbf{x}, \tilde{\mathbf{x}}; \boldsymbol{\theta})$ as follows:

$$\hat{p}_{Y|X\tilde{X}}(y|\mathbf{x}, \tilde{\mathbf{x}}; \boldsymbol{\theta}) = \frac{\hat{p}_{Y|\tilde{X}}(y|\tilde{\mathbf{x}}; \boldsymbol{\theta}) \exp h_{\boldsymbol{\theta}}(\mathbf{x}, \tilde{\mathbf{x}}, y)}{\sum_{\hat{y} \in \mathcal{Y}} \hat{p}_{Y|\tilde{X}}(\hat{y}|\tilde{\mathbf{x}}; \boldsymbol{\theta}) \exp h_{\boldsymbol{\theta}}(\mathbf{x}, \tilde{\mathbf{x}}, \hat{y})}. \tag{23}$$

Combining Eq. (23) with Eq. (22), we have the following:

$$
\begin{aligned}
\hat{\mathcal{W}}(\mathbf{x}, y|\tilde{\mathbf{x}}; \boldsymbol{\theta}) &= \log \frac{\exp h_{\boldsymbol{\theta}}(\mathbf{x}, \tilde{\mathbf{x}}, y)}{\sum_{\hat{y} \in \mathcal{Y}} \hat{p}_{Y|\tilde{X}}(\hat{y}|\tilde{\mathbf{x}}; \boldsymbol{\theta}) \exp h_{\boldsymbol{\theta}}(\mathbf{x}, \tilde{\mathbf{x}}, \hat{y})} \\
&= \log \frac{\exp h_{\boldsymbol{\theta}}(\mathbf{x}, \tilde{\mathbf{x}}, y)}{\mathbb{E}_{\hat{y} \sim \mathbb{P}_Y} [\eta(\tilde{\mathbf{x}}, \hat{y}) \exp h_{\boldsymbol{\theta}}(\mathbf{x}, \tilde{\mathbf{x}}, \hat{y})]} \\
&\approx \log \frac{\exp h_{\boldsymbol{\theta}}(\mathbf{x}, \tilde{\mathbf{x}}, y)}{\sum_{j=1}^{K+L} \eta(\tilde{\mathbf{x}}, \hat{y}) \exp h_{\boldsymbol{\theta}}(\mathbf{x}, \tilde{\mathbf{x}}, y_j)} + \log(K + L),
\end{aligned} \tag{24}
$$

where $\eta(\tilde{\mathbf{x}}, \hat{y}) = \hat{p}_{Y|\tilde{X}}(\hat{y}|\tilde{\mathbf{x}}; \boldsymbol{\theta})/p_Y(\hat{y})$. We note that it is suffice to follow Eq. (6) to derive the last step of Eq. (24), where, as suggested by Theorem 1, both negative labels and ID labels are leveraged for the Monte-Carlo estimation of the expectation. Recalling that, according to Eq. (19), $\eta(\tilde{\mathbf{x}}, \hat{y}) = \exp \hat{\mathcal{W}}(\tilde{\mathbf{x}}, \hat{y}; \boldsymbol{\theta})$, connecting Eq. (24) to Eq. (18) results in reformulating the estimated energy function $E_{\boldsymbol{\theta}}(\mathbf{x})$ in Eq. (6) with $\alpha = 1$ as follows:

$$
\begin{aligned}
E_{\boldsymbol{\theta}}(\mathbf{x}) &= \log \sum_{y \in \mathcal{Y}_I} \exp \left[ \hat{\mathcal{W}}(\tilde{\mathbf{x}}, y; \boldsymbol{\theta}) + \hat{\mathcal{W}}(\mathbf{x}, y|\tilde{\mathbf{x}}; \boldsymbol{\theta}) \right] \\
&= \log \sum_{y \in \mathcal{Y}_I} \frac{\exp \left[ \hat{\mathcal{W}}(\tilde{\mathbf{x}}, y; \boldsymbol{\theta}) + h_{\boldsymbol{\theta}}(\mathbf{x}, \tilde{\mathbf{x}}, y) \right]}{\mathbb{E}_{\hat{y} \sim \mathbb{P}_Y} \left[ \exp \left[ \hat{\mathcal{W}}(\tilde{\mathbf{x}}, \hat{y}; \boldsymbol{\theta}) + h_{\boldsymbol{\theta}}(\mathbf{x}, \tilde{\mathbf{x}}, \hat{y}) \right] \right]} \\
&\approx \log S_{\text{ours}}(\mathbf{x}; \boldsymbol{\theta}) + \log(K + L),
\end{aligned} \tag{25}
$$

where, inspired by Tsai et al. [63], we define $h_{\boldsymbol{\theta}}(\mathbf{x}, \tilde{\mathbf{x}}, y) \triangleq \mathbf{r}^\top [\beta \tilde{\mathbf{z}} + (1 - \beta)\mathbf{z}]/\kappa$ with $\beta \in (0, 1)$ and $\kappa > 0$ as two hyper-parameters, and

$$S_{\text{ours}}(\mathbf{x}; \boldsymbol{\theta}) \triangleq \sum_{y \in \mathcal{Y}_I} \frac{\exp \left[ \Lambda(\tilde{\mathbf{x}}, y; \boldsymbol{\theta}) + h_{\boldsymbol{\theta}}(\mathbf{x}, \tilde{\mathbf{x}}, y) \right]}{\sum_{j=1}^{K+L} \exp \left[ \Lambda(\tilde{\mathbf{x}}, y_j; \boldsymbol{\theta}) + h_{\boldsymbol{\theta}}(\mathbf{x}, \tilde{\mathbf{x}}, y_j) \right]}. \tag{26}$$

Similarly, we present the following theorem to reveal the provable guarantee of how $S_{\text{ours}}(\mathbf{x}; \boldsymbol{\theta})$ correctly recovers the true energy function $E(\mathbf{x})$ in Eq. (4).

**Theorem 4.** *Let $h_{\boldsymbol{\theta}}(\mathbf{x}, \tilde{\mathbf{x}}, y) \triangleq \mathbf{r}^\top [\beta \tilde{\mathbf{z}} + (1 - \beta)\mathbf{z}]/\kappa$, $h_{\boldsymbol{\theta}}(\tilde{\mathbf{x}}, y) = \tilde{\mathbf{z}}^\top \mathbf{r}/\tau$, $\alpha = 1$ and $N = L + K/T$. If we, following prior works [56, 38], assume that $h_{\boldsymbol{\theta}}(\tilde{\mathbf{x}}, y) = \log \frac{p_{Y|\tilde{X}}(y|\tilde{\mathbf{x}})}{p_Y(y)} + c(\tilde{\mathbf{x}})$ with $c(\tilde{\mathbf{x}})$ as a constant term depending on $\tilde{\mathbf{x}}$, and that $h_{\boldsymbol{\theta}}(\mathbf{x}, \tilde{\mathbf{x}}, y) = \log \frac{p_{Y|X\tilde{X}}(y|\mathbf{x}, \tilde{\mathbf{x}})}{p_{Y|\tilde{X}}(y|\tilde{\mathbf{x}})} + c(\mathbf{x}, \tilde{\mathbf{x}})$ with $c(\mathbf{x}, \tilde{\mathbf{x}})$ as a constant term depending on $\mathbf{x}$ and $\tilde{\mathbf{x}}$, then we have the following[10]:*

$$E(\mathbf{x}) = \lim_{N \to +\infty} E_{\boldsymbol{\theta}}(\mathbf{x}) = \lim_{\substack{T \to +\infty \\ L/T \to +\infty}} \log(K + L) S_{\text{ours}}(\mathbf{x}; \boldsymbol{\theta}). \tag{27}$$

**Remark.** Comparing Eq. (25) with Eq. (10) and Eq. (14), one can find that $S_{\text{ours}}(\mathbf{x}; \boldsymbol{\theta})$ capture at most $\log(K + L)$ nats of the true $E(\mathbf{x})$, which is strictly larger than $\log K$ in $S_{\text{MCM}}(\mathbf{x}; \boldsymbol{\theta})$ and $\log(K + L/T)$ in $S_{\text{Neglabel}}(\mathbf{x}; \boldsymbol{\theta})$. Although this upper bound, i.e., $\log(K + L)$, can be achieved by $S_{\text{Neglabel}}(\mathbf{x}; \boldsymbol{\theta})$ in Eq. (14) by either 1) introducing $TL$ negative labels or 2) fixing $T = 1$, we note that 1) NegLabel [26] has empirically observed the degeneration of OOD detection performance caused by excessive negative labels, and that 2) decreasing $T$ can be in conflict with Theorem 2 where the recovery of $E(\mathbf{x})$ with $S_{\text{NegLabel}}(\mathbf{x}; \boldsymbol{\theta})$ explicitly requires $T$ to be sufficiently large.

---

[10]We detail the derivation in Appendix F

Table 1: OOD detection results on the ImageNet-1K dataset. ↑ indicates larger values are better and vice versa. The best results in the last two columns are shown in bold.

| Method | iNaturalist | | SUN | | Places | | Textures | | Average | |
|---|---|---|---|---|---|---|---|---|---|---|
| | AUROC↑ | FPR95↓ | AUROC↑ | FPR95↓ | AUROC↑ | FPR95↓ | AUROC↑ | FPR95↓ | AUROC↑ | FPR95↓ |
| **Methods requiring training (or fine-tuning)** | | | | | | | | | | |
| MSP | 87.44 | 58.36 | 79.73 | 73.72 | 79.67 | 74.41 | 79.69 | 71.93 | 81.63 | 69.61 |
| ODIN | 94.65 | 30.22 | 87.17 | 54.04 | 85.54 | 55.06 | 87.85 | 51.67 | 88.80 | 47.75 |
| Energy | 95.33 | 26.12 | 92.66 | 35.97 | 91.41 | 39.87 | 86.76 | 57.61 | 91.54 | 39.89 |
| GradNorm | 72.56 | 81.50 | 72.86 | 82.00 | 73.70 | 80.41 | 70.26 | 79.36 | 72.35 | 80.82 |
| ViM | 93.16 | 32.19 | 87.19 | 54.01 | 83.75 | 60.67 | 87.18 | 53.94 | 87.82 | 50.20 |
| KNN | 94.52 | 29.17 | 92.67 | 35.62 | 91.02 | 39.61 | 85.67 | 64.35 | 90.97 | 42.19 |
| VOS | 94.62 | 28.99 | 92.57 | 36.88 | 91.23 | 38.39 | 86.33 | 61.02 | 91.19 | 41.32 |
| NPOS | 96.19 | 16.58 | 90.44 | 43.77 | 89.44 | 45.27 | 88.90 | 46.12 | 91.22 | 37.93 |
| LSN | 95.83 | 21.56 | 94.35 | 26.32 | 91.25 | 34.48 | 90.42 | 38.54 | 92.96 | 30.22 |
| CLIPN | 95.27 | 23.94 | 93.93 | 26.17 | 92.28 | 33.45 | 90.93 | 40.83 | 93.10 | 31.10 |
| LoCoOp | 96.86 | 16.05 | 95.07 | 23.44 | 91.98 | 32.87 | 90.19 | 42.28 | 93.52 | 28.66 |
| LAPT | 99.63 | 1.16 | 96.01 | 19.12 | 92.01 | 33.01 | 91.06 | 40.32 | 94.68 | 23.40 |
| NegPro | 98.73 | 6.32 | 95.55 | 22.89 | 93.34 | 27.60 | 91.60 | 35.21 | 94.81 | 23.01 |
| HFTT | 93.27 | 27.44 | 95.28 | 19.24 | 90.26 | 43.54 | 88.23 | 43.08 | 91.76 | 33.33 |
| **Zero-Shot Training-free Methods** | | | | | | | | | | |
| ZOC | 86.09 | 87.30 | 81.20 | 81.51 | 83.39 | 73.06 | 76.46 | 98.90 | 81.79 | 85.19 |
| MCM | 94.59 | 32.20 | 92.25 | 38.80 | 90.31 | 46.20 | 86.12 | 58.50 | 90.82 | 43.93 |
| NegLabel | 99.49 | 1.91 | 95.49 | 20.53 | 91.64 | 35.59 | 90.22 | 43.56 | 94.21 | 25.40 |
| Ours (Median) | 99.70 | 1.04 | 96.16 | 16.06 | 93.37 | 26.92 | 91.01 | 40.78 | **95.07** | **21.20** |
| Ours (Mean) | 99.64 | 1.04 | 96.32 | 18.45 | 95.81 | 31.15 | 92.15 | 38.79 | **96.00** | **22.36** |

# 6 Experiments

**Evaluation Metrics**. The performance of OOD detection is evaluated via two widely used metrics: 1) the false positive rate of OOD data is measured when the true positive rate of ID data reaches 95% (FPR95); 2) the area under the receiver operating characteristic curve (AUROC) is computed to quantify the probability of the ID case receiving a higher score than the OOD case.

**Baseline Methods**. We compare our method with MSP [20], ODIN [34], Energy [36], KNN [61], Gradnorm [24], Vim [66], VOS [12], NPOS [62], ZOC [15], CLIPN [68], LoCoOp [43], LSN [46], LAPT [76], NegPro [33], HFTT [32], MCM [42], NegLabel [26] and AdaNeg [75].

**Implementation Details**. Unless otherwise specified, we employ the CLIP ViT-B/16 model as the pre-trained VLM. We use the same NegMining algorithm as NegLabel [26] to extract top 15% dissimilar words to ID labels from WordNet as negative labels, followed by separating the negative labels into $T = 10$ groups for OOD scoring. Following NegLabel [26], we adopt the text prompt of 'The nice <label>.'. We apply the random cropping augmentation on each test-time image $\mathbf{x}$ with the scale range $(\lambda, 1.0)$ to produce the sub-view $\tilde{\mathbf{x}}$, followed by resizing $\tilde{\mathbf{x}}$ to $224 \times 224$. Regarding hyper-parameters in main results, we set $\tau = 0.02$, $\kappa = 0.08$, $\lambda = 0.55$, $\alpha = 0.8$ and $\beta = 0.3$. To reduce variance of random cropping, the final OOD scoring function is averaged over $V = 2$ randomly cropped sub-views. All experiments are conducted with a single Tesla A100 GPU. Source codes for reproduction can be found in Supplementary Materials.

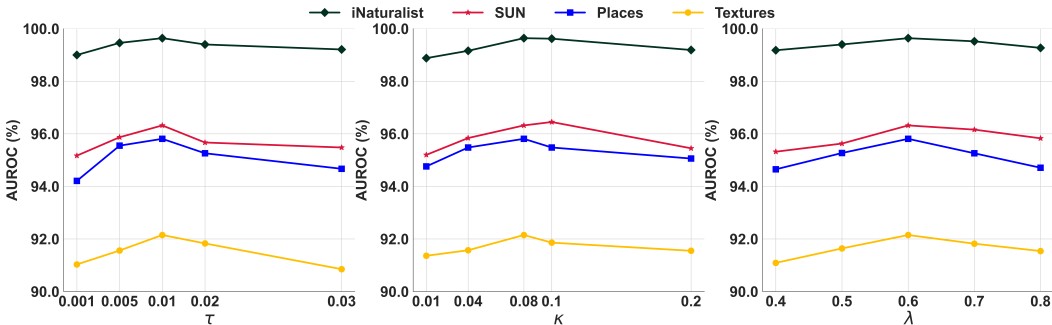

Figure 1: Ablation study on ImageNet-1K w.r.t hyper-parameters $\tau$ (left), $\kappa$ (middle) and $\lambda$ (right)

Table 2: Evaluation on domain-generalizable OOD detection. ↑ indicates larger values are better and vice versa. The best results in the last two columns are shown in bold per ID dataset.

| ID Dataset | Method | iNaturalist | | SUN | | Places | | Textures | | Average | |
|---|---|---|---|---|---|---|---|---|---|---|---|
| | | AUROC↑ | FPR95↓ | AUROC↑ | FPR95↓ | AUROC↑ | FPR95↓ | AUROC↑ | FPR95↓ | AUROC↑ | FPR95↓ |
| | MCM | 87.74 | 63.06 | 85.35 | 67.24 | 81.19 | 70.64 | 74.77 | 79.59 | 82.26 | 70.13 |
| ImageNet-S | NegLabel | 99.34 | 2.24 | 94.93 | 22.73 | 90.78 | 38.62 | 89.29 | 46.10 | 93.59 | 27.42 |
| | Ours | 99.51 | 1.62 | 96.02 | 20.17 | 93.89 | 34.69 | 91.35 | 42.94 | **95.52** | **25.11** |
| | MCM | 79.50 | 76.85 | 76.19 | 79.78 | 70.95 | 80.51 | 61.98 | 86.37 | 80.88 | 72.16 |
| ImageNet-A | NegLabel | 98.80 | 4.09 | 89.83 | 44.38 | 82.88 | 60.10 | 80.25 | 64.34 | 87.94 | 43.23 |
| | Ours | 99.16 | 3.58 | 91.64 | 39.63 | 86.25 | 55.64 | 87.43 | 58.76 | **91.23** | **39.40** |

## 6.1 Main Results

We conduct experiments on the ImageNet dataset, demonstrating the scalability of our method. Specifically, we inherit the setup from prior work [42, 26, 75], where the ID dataset is ImageNet-1K [9] and OOD datasets include iNaturalist [64], SUN [70], Places365 [80], and Textures [7]. At test time, all images are resized to 224×224. Table 1 presents the performance of our approach and existing competitive baselines, where the proposed approach significantly outperforms existing methods. Specifically, advanced post-hoc methods generally perform better than learning-based methods especially when the SUN dataset acts as the OOD data without requiring additional training. Besides, compared with the state-of-the-art NegLabel, our method reveals 3.16% and 2.21% averaged improvement w.r.t FPR95 and AUROC on the ImageNet dataset. For advanced works, i.e., NegLabel+AdaNeg, that additionally consider visual negative proxies in OOD scoring, our improved version, i.e., Ours+AdaNeg, performs better on all four OOD datasets.

## 6.2 Ablation Study

We analyze the hyper-parameters most essential to our algorithmic design, including the minimum crop scale $\lambda$ and two scaling temperatures $\tau$ and $\kappa$. The corresponding results are plotted in Figure 1. On the one hand, having a large or small value of the two scaling temperatures does not necessarily improve the OOD detection performance while our method consistently outperforms the state-of-the-art NegLabel when the value of $\tau$ and $\kappa$ varies from 0.05 to 0.02 and from 0.04 to 0.1 respectively. On the other hand, it can be found that the aggressive cropping strategy, which corresponding to that the value $\lambda$ is small, can deteriorate the OOD detection. We suspect that this is because aggressive cropping may hurt the semantics of the original image.

## 6.3 Extensions

**Domain-generalizable OOD Detection.** We consider domain generalizable OOD detection scenarios, where domain shifts occur in ID data. With ImageNet-1K as the ID data, we, following NegLabel [26], consider ImageNet-S [65] and ImageNet-A [23] as ID data receptively. The performance gain in Table 4 implies the more robustness of our method to domain shift.

Table 3: OOD detection results on the ImageNet-1K with various learned prompts, i.e., NegPro [33] and LAPT [76], respectively. Following Zhang & Zhang [75], the performance is measured by FPR95 ↓. The best results are shown in bold.

| Method | iNaturalist | | SUN | | Places | | Textures | | Average | |
|---|---|---|---|---|---|---|---|---|---|---|
| | NegPro | LAPT | NegPro | LAPT | NegPro | LAPT | NegPro | LAPT | NegPro | LAPT |
| NegLabel+AdaNeg | 3.87 | 0.58 | 11.35 | 9.98 | 25.45 | 30.47 | 29.79 | 25.25 | 17.62 | 16.32 |
| Ours+AdaNeg | 4.16 | 0.63 | 9.47 | 8.39 | 23.79 | 25.64 | 26.42 | 25.76 | **15.96** | **15.11** |

**OOD Detection with Learned Prompt.** While this paper, following Neglabel [26], to use a pre-defined prompts for ID label, we show that our method can be made stronger with the mostly recent technology of prompt learning. Empirically, we compare the results in Table 5 by using the prompts learned by either Negpro [33] or LAPT [76].

# 7 Conclusion

This paper presents a information-theoretic framework to characterizes and unifies the theoretical understanding of post-hoc OOD detection with pre-trained VLMs. In particular, by modeling the ID data with an energy-based model with the PMI between the input image and each ID label as energy functions, We show that representative CLIP-based post-hoc OOD detection methods implicitly work as stochastic Monte Carlo estimations of PMI for density estimation. Motivated by the *divide-and-conquer* philosophy, we decompose the original PMI into a sum of conditional and unconditional PMI terms to facilitate OOD detection, which demonstrates both theoretical and empirical superiority.

## Acknowledgement

This work is supported by Australian Research Council Discovery Early Career Researcher Award (DE250100363) and Australian Laureate Fellowship (FL190100149).

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

# A  Related Works on Traditional OOD Detection

On the theoretical side, there are various attempts to explore the theoretical understanding of OOD detection. Fang et al. [16, 17] study the generalization of OOD detection by PAC learning and find a necessary condition for the learnability of OOD detection. Morteza & Li [44] provides a provable understanding of the OOD detection result by modelling the feature embedding space as a mixture of multivariate Gaussian distributions. Du et al. [14] studies the impact of ID labels on OOD detection.

On the practical side, the popularity of OOD detection is motivated by the empirical observation [45] that neural networks tend to be over-confident in OOD data despite the remarkable achievement in applications [83, 48, 82, 3, 85, 84, 49, 50, 4, 81, 86, 53, 54]. One line of work performs OOD detection by devising post-hoc scoring functions, including confidence-based methods [37, 22, 78], energy-based methods [36, 67], distance-based approaches [61, 31, 59, 74, 1, 52, 77, 55], gradient-based approaches [24], generative approaches [73, 18, 35], and Bayesian approaches [40]. Another line of work addresses OOD detection by fine-tuning a pre-trained discrimination model with training-time regularizations that help the model learn ID/OOD discrepancy following the guideline of outlier exposure [21]. For instance, the discriminative model is regularized to produce lower confidence [30, 39] or higher energy [36] for outlier points. More recently, some works consider a more practical but challenging scenario where auxiliary outliers are contaminated with unlabelled ID counterparts. WOOD [27] formulates learning with noisy OOD data as a constrained optimization problem while SAL [13] separates candidate outliers from the unlabeled data and trains a binary classifier using the candidate outliers and the labelled ID data.

# B  Proof of Theorem 1

As a reminder, Theorem 1 is stated as follows:

**Theorem 1.** *Let $h_{\boldsymbol{\theta}}(\mathbf{x}, y) = \mathbf{z}^{\top}\mathbf{r}/\tau$ and $N = K$. If we, following prior works [56, 47], assume that $h_{\boldsymbol{\theta}}(\mathbf{x}, y) = \log \frac{p_{Y|X}(y|\mathbf{x})}{p_Y(y)} + c(\mathbf{x})$ with $c(\mathbf{x})$ as a constant term depending on $\mathbf{x}$, in the extreme case where $\alpha \to +\infty$, we then have the following:*

$$\lim_{\alpha \to +\infty} E(\mathbf{x}) = \lim_{\alpha \to +\infty} E_{\boldsymbol{\theta}}(\mathbf{x}) = \lim_{N \to +\infty} \log N S_{MCM}(\mathbf{x}; \boldsymbol{\theta}). \tag{28}$$

*Proof.* **Step 1:**

$$
\begin{aligned}
\lim_{\alpha \to +\infty} E_{\boldsymbol{\theta}}(\mathbf{x}) &= \lim_{\alpha \to +\infty} \frac{1}{\alpha} \log \sum_{y \in \mathcal{Y}_{\mathrm{I}}} \exp \left[ \alpha \log \frac{\hat{p}_{Y|X}(y|\mathbf{x}; \boldsymbol{\theta})}{p_Y(y)} \right] \\
&= \lim_{\alpha \to +\infty} \frac{1}{\alpha} \log \sum_{y \in \mathcal{Y}_{\mathrm{I}}} \exp \left[ \alpha \log \frac{\exp h_{\boldsymbol{\theta}}(\mathbf{x}, y)}{\mathbb{E}_{\hat{y} \sim \mathbb{P}_Y} \left[ \exp h_{\boldsymbol{\theta}}(\mathbf{x}, \hat{y}) \right]} \right] \\
&= \lim_{\alpha \to +\infty} \frac{1}{\alpha} \log \sum_{y \in \mathcal{Y}_{\mathrm{I}}} \exp \left[ \alpha \log \frac{\frac{p_{Y|X}(y|\mathbf{x})}{p_Y(y)} \exp c(\mathbf{x})}{\mathbb{E}_{\hat{y} \sim \mathbb{P}_Y} \left[ \frac{p_{Y|X}(\hat{y}|\mathbf{x})}{p_Y(\hat{y})} \exp c(\mathbf{x}) \right]} \right] \\
&= \lim_{\alpha \to +\infty} \frac{1}{\alpha} \log \sum_{y \in \mathcal{Y}_{\mathrm{I}}} \exp \left[ \alpha \log \frac{p_{Y|X}(y|\mathbf{x})}{p_Y(y)} \right] = \lim_{\alpha \to +\infty} E(\mathbf{x}),
\end{aligned}
\tag{29}
$$

where the penultimate step of Eq. (29) is derived based on the fact that

$$\mathbb{E}_{\hat{y}\sim\mathbb{P}_Y}\left[\frac{p_{Y|X}(y|\mathbf{x})}{p_Y(\hat{y})}\right] = \sum_{\hat{y}\in\mathcal{Y}}p_Y(\hat{y})\frac{p_{Y|X}(\hat{y}|\mathbf{x})}{p_Y(\hat{y})} = \sum_{\hat{y}\in\mathcal{Y}}p_{Y|X}(\hat{y}|\mathbf{x}) = 1. \tag{30}$$

**Step 2:**

Given that $h_{\boldsymbol{\theta}}(\mathbf{x},y) = \mathbf{z}^\top\mathbf{r}_j/\tau$, and that, as implied by the law of large numbers, $\lim_{N\to+\infty}\frac{1}{N}\sum_{j=1}^N\exp h_{\boldsymbol{\theta}}(\mathbf{x},y_j) = \mathbb{E}_{\hat{y}\sim\mathbb{P}_Y}[\exp h_{\boldsymbol{\theta}}(\mathbf{x},\hat{y})]$ (for all $y_j\in\mathcal{Y}$), we have the following:

$$
\begin{aligned}
\lim_{N\to+\infty}\log NS_{\mathrm{MCM}}(\mathbf{x};\boldsymbol{\theta}) &= \lim_{N\to+\infty}\log\max_{y\in\mathcal{Y}_{\mathrm{I}}}\frac{N\exp(\mathbf{z}^\top\mathbf{r}/\tau)}{\sum_{j=1}^N\exp(\mathbf{z}^\top\mathbf{r}_j/\tau)}\\
&= \log\max_{y\in\mathcal{Y}_{\mathrm{I}}}\exp(\mathbf{z}^\top\mathbf{r}/\tau) - \lim_{N\to+\infty}\log\frac{1}{N}\sum_{j=1}^N\exp(\mathbf{z}^\top\mathbf{r}_j/\tau)\\
&= \log\max_{y\in\mathcal{Y}_{\mathrm{I}}}\exp(\mathbf{z}^\top\mathbf{r}/\tau) - \log\lim_{N\to+\infty}\frac{1}{N}\sum_{j=1}^N\exp(\mathbf{z}^\top\mathbf{r}_j/\tau)\\
&= \log\max_{y\in\mathcal{Y}_{\mathrm{I}}}\exp(\mathbf{z}^\top\mathbf{r}/\tau) - \log\mathbb{E}_{\hat{y}\sim\mathbb{P}_Y}\left[\exp(\mathbf{z}^\top\mathbf{r}/\tau)\right]\\
&= \log\max_{y\in\mathcal{Y}_{\mathrm{I}}}\frac{\exp(\mathbf{z}^\top\mathbf{r}/\tau)}{\mathbb{E}_{\hat{y}\sim\mathbb{P}_Y}\left[\exp(\mathbf{z}^\top\mathbf{r}/\tau)\right]}\\
&= \max_{y\in\mathcal{Y}_{\mathrm{I}}}\log\frac{\exp(\mathbf{z}^\top\mathbf{r}/\tau)}{\mathbb{E}_{\hat{y}\sim\mathbb{P}_Y}\left[\exp(\mathbf{z}^\top\mathbf{r}/\tau)\right]}\\
&= \lim_{\alpha\to+\infty}\frac{1}{\alpha}\log\sum_{y\in\mathcal{Y}_{\mathrm{I}}}\exp\left[\alpha\log\frac{\exp(\mathbf{z}^\top\mathbf{r}/\tau)}{\mathbb{E}_{\hat{y}\sim\mathbb{P}_Y}\left[\exp(\mathbf{z}^\top\mathbf{r}/\tau)\right]}\right]\\
&= \lim_{\alpha\to+\infty}\frac{1}{\alpha}\log\sum_{y\in\mathcal{Y}_{\mathrm{I}}}\exp\left[\alpha\log\frac{\exp h_{\boldsymbol{\theta}}(\mathbf{x},y)}{\mathbb{E}_{\hat{y}\sim\mathbb{P}_Y}\left[\exp h_{\boldsymbol{\theta}}(\mathbf{x},\hat{y})\right]}\right]\\
&= \lim_{\alpha\to+\infty}E_{\boldsymbol{\theta}}(\mathbf{x})
\end{aligned}
\tag{31}
$$

Step 1 and Step 2 imply this result.

$\square$

# C    Justification of $\hat{p}_{Y|X\mathbf{Y}_N}(y|\mathbf{x},\hat{\mathbf{y}}_N;\boldsymbol{\theta})$

As a reminer, $\hat{p}_{Y|X\mathbf{Y}_N}(y|\mathbf{x},\hat{\mathbf{y}}_N;\boldsymbol{\theta})$ is given as follows:

$$\hat{p}_{Y|X\mathbf{Y}_N}(y|\mathbf{x},\hat{\mathbf{y}}_N;\boldsymbol{\theta}) = \frac{p_Y(y)\exp h_{\boldsymbol{\theta}}(\mathbf{x},y)}{\exp h_{\boldsymbol{\theta}}(\mathbf{x},y) + \sum_{j=1}^N\exp h_{\boldsymbol{\theta}}(\mathbf{x},\hat{y}_j)}, \tag{32}$$

$$\sum_{y\in\mathcal{Y}}\hat{p}_{Y|\tilde{X}}(y|\tilde{\mathbf{x}};\boldsymbol{\theta}) = \sum_{y\in\mathcal{Y}}\mathbb{E}_{\hat{\mathbf{y}}_{N-1}\sim\mathbb{P}_Y^{N-1}}\left[\hat{p}_{Y|X\mathbf{Y}_{N-1}}(y|\mathbf{x},\hat{\mathbf{y}}_{N-1};\boldsymbol{\theta})\right]$$

$$= \sum_{y\in\mathcal{Y}}\mathbb{E}_{\hat{\mathbf{y}}_{N-1}\sim\mathbb{P}_Y^{N-1}}\left[N\frac{p_Y(y)\exp h_{\boldsymbol{\theta}}(\tilde{\mathbf{x}},y)}{\exp h_{\boldsymbol{\theta}}(\mathbf{x},y)+\sum_{j=1}^{N-1}\exp h_{\boldsymbol{\theta}}(\mathbf{x},\hat{y}_j)}\right]$$

$$= \sum_{y\in\mathcal{Y}}p_Y(y)\mathbb{E}_{\hat{\mathbf{y}}_{N-1}\sim\mathbb{P}_Y^{N-1}}\left[N\frac{\exp h_{\boldsymbol{\theta}}(\tilde{\mathbf{x}},y)}{\exp h_{\boldsymbol{\theta}}(\mathbf{x},y)+\sum_{j=1}^{N-1}\exp h_{\boldsymbol{\theta}}(\mathbf{x},\hat{y}_j)}\right]$$

$$= \mathbb{E}_{y\sim\mathbb{P}_Y}\mathbb{E}_{\hat{\mathbf{y}}_{N-1}\sim\mathbb{P}_Y^{N-1}}\left[N\frac{\exp h_{\boldsymbol{\theta}}(\tilde{\mathbf{x}},y)}{\exp h_{\boldsymbol{\theta}}(\mathbf{x},y)+\sum_{j=1}^{N-1}\exp h_{\boldsymbol{\theta}}(\mathbf{x},\hat{y}_j)}\right] \quad (33)$$

$$= N\mathbb{E}_{\hat{\mathbf{y}}_N\sim\mathbb{P}_Y^N}\left[\frac{\exp h_{\boldsymbol{\theta}}(\tilde{\mathbf{x}},\hat{y}_N)}{\sum_{j=1}^{N}\exp h_{\boldsymbol{\theta}}(\mathbf{x},\hat{y}_j)}\right]$$

$$= \sum_{i=1}^{N}\mathbb{E}_{\hat{\mathbf{y}}_N\sim\mathbb{P}_Y^N}\left[\frac{\exp h_{\boldsymbol{\theta}}(\tilde{\mathbf{x}},\hat{y}_i)}{\sum_{j=1}^{N}\exp h_{\boldsymbol{\theta}}(\mathbf{x},\hat{y}_j)}\right]$$

$$= \mathbb{E}_{\hat{\mathbf{y}}_N\sim\mathbb{P}_Y^N}\left[\frac{\sum_{i=1}^{N}\exp h_{\boldsymbol{\theta}}(\tilde{\mathbf{x}},\hat{y}_i)}{\sum_{j=1}^{N}\exp h_{\boldsymbol{\theta}}(\mathbf{x},\hat{y}_j)}\right] = 1$$

Please refer to Section 5.2 in Cremer et al. [8] for more details of the derivation.

## D Proof of Theorem 2

As a reminder, Theorem 2 is stated as follows:

**Theorem 2.** *Let $h_{\boldsymbol{\theta}}(\mathbf{x},y) = \mathbf{z}^\top\mathbf{r}/\tau$, $\alpha = 1$, and $N = K+L/T$. If we, following prior works [56, 47], assume that $h_{\boldsymbol{\theta}}(\mathbf{x},y) = \log\frac{p_{Y|X}(y|\mathbf{x})}{p_Y(y)} + c(\mathbf{x})$ with $c(\mathbf{x})$ as a constant term depending on $\mathbf{x}$, we then have the following:*

$$E(\mathbf{x}) = \lim_{N\to+\infty}E_{\boldsymbol{\theta}}(\mathbf{x}) = \lim_{\substack{T\to+\infty\\L/T\to+\infty}}\log(K+L/T)S_{NegLabel}(\mathbf{x};\boldsymbol{\theta}). \quad (34)$$

*Proof.* Given that $\alpha = 1$, we have

$$E(\mathbf{x}) = \log\sum_{y\in\mathcal{Y}_\mathrm{I}}\exp\left[\mathcal{W}(x,y)\right] = \log\sum_{y\in\mathcal{Y}_\mathrm{I}}\exp\left[\log\frac{p_{Y|X}(y|\mathbf{x})}{p_Y(y)}\right] = \log\sum_{y\in\mathcal{Y}_\mathrm{I}}\frac{p_{Y|X}(y|\mathbf{x})}{p_Y(y)}. \quad (35)$$

$$E_{\boldsymbol{\theta}}(\mathbf{x}) = \log\sum_{y\in\mathcal{Y}_\mathrm{I}}\exp\left[\log\hat{\mathcal{W}}(\mathbf{x},y)\right] = \log\sum_{y\in\mathcal{Y}_\mathrm{I}}\frac{\hat{p}_{Y|X}(y|\mathbf{x};\boldsymbol{\theta})}{p_Y(y)} \quad (36)$$

**Step 1:**

Given that $h_{\boldsymbol{\theta}}(\mathbf{x},y) = \log\frac{p_{Y|X}(y|\mathbf{x})}{p_Y(y)} + c(\mathbf{x})$, when $N\to+\infty$, the Law of Large Numbers implies the following

$$\lim_{N\to+\infty}\frac{1}{N}\left[\exp h_{\boldsymbol{\theta}}(\mathbf{x},y)+\sum_{j=1}^{N-1}\exp h_{\boldsymbol{\theta}}(\mathbf{x},\hat{y}_j)\right]$$

$$= \lim_{N\to+\infty}\frac{1}{N}\exp h_{\boldsymbol{\theta}}(\mathbf{x},y)+\lim_{N\to+\infty}\frac{N-1}{N}\lim_{N\to+\infty}\frac{1}{N-1}\sum_{j=1}^{N-1}\exp h_{\boldsymbol{\theta}}(\mathbf{x},\hat{y}_j) \quad (37)$$

$$= \mathbb{E}_{\hat{y}\sim\mathbb{P}_Y}\left[\exp h_{\boldsymbol{\theta}}(\mathbf{x},\hat{y})\right]$$

$$= \mathbb{E}_{\hat{y}\sim\mathbb{P}_Y}\left[\frac{p_{Y|X}(\hat{y}|\mathbf{x})}{p_Y(\hat{y})}\exp c(\mathbf{x})\right] = \exp c(\mathbf{x})$$

**Step 2:**

This implies that

$$
\lim_{N \to +\infty} E_{\boldsymbol{\theta}}(\mathbf{x}) = \lim_{N \to +\infty} \log \sum_{y \in \mathcal{Y}_{\mathrm{I}}} \mathbb{E}_{\hat{\mathbf{y}}_{N-1} \sim \mathbb{P}_Y^{N-1}} \left[ \frac{N \cdot \exp h_{\boldsymbol{\theta}}(\mathbf{x}, y)}{\exp h_{\boldsymbol{\theta}}(\mathbf{x}, y) + \sum_{j=1}^{N-1} \exp h_{\boldsymbol{\theta}}(\mathbf{x}, \hat{y}_j)} \right]
$$

$$
= \lim_{N \to +\infty} \log \sum_{y \in \mathcal{Y}_{\mathrm{I}}} \mathbb{E}_{\hat{\mathbf{y}}_{N-1} \sim \mathbb{P}_Y^{N-1}} \left[ \frac{p_{Y|X}(y|\mathbf{x})}{p_Y(y)} \right] \tag{38}
$$

$$
= \log \sum_{y \in \mathcal{Y}_{\mathrm{I}}} \frac{p_{Y|X}(y|\mathbf{x})}{p_Y(y)} = E(\mathbf{x})
$$

**Step 3:** Let $\hat{y}_i^{(t)}$ be the $i$-th sample drawn at the $t$-th round, the law of large numbers implies

$$
\frac{\hat{p}_{Y|X}(y|\mathbf{x}; \boldsymbol{\theta})}{p_Y(y)} = \mathbb{E}_{\hat{\mathbf{y}}_{N-1} \sim \mathbb{P}_Y^{N-1}} \left[ \frac{N \cdot \exp h_{\boldsymbol{\theta}}(\mathbf{x}, y)}{\exp h_{\boldsymbol{\theta}}(\mathbf{x}, y) + \sum_{j=1}^{N-1} \exp h_{\boldsymbol{\theta}}(\mathbf{x}, \hat{y}_j)} \right]
$$

$$
= \lim_{T \to +\infty} \frac{1}{T} \sum_{i=1}^{T} \frac{N \cdot \exp h_{\boldsymbol{\theta}}(\mathbf{x}, y)}{\exp h_{\boldsymbol{\theta}}(\mathbf{x}, y) + \sum_{j=1}^{N-1} \exp h_{\boldsymbol{\theta}}(\mathbf{x}, \hat{y}_j^{(t)})} \tag{39}
$$

$$
= \lim_{T \to +\infty} \frac{N}{T} \sum_{i=1}^{T} \hat{L}(x, y, t),
$$

where, for each $t = 1, \ldots, T$,

$$
\hat{L}(x, y, t) = \frac{\exp h_{\boldsymbol{\theta}}(\mathbf{x}, y)}{\exp h_{\boldsymbol{\theta}}(\mathbf{x}, y) + \sum_{j=1}^{N-1} \exp h_{\boldsymbol{\theta}}(\mathbf{x}, \hat{y}_j^{(t)})} \tag{40}
$$

As $\hat{\mathbf{y}}_{N-1}^{(t)} = (\hat{y}_i^{(t)})_{i=1}^{N-1}$ is sampled from $\mathbb{P}_Y^{N-1}$ i.i.d, we can build $\hat{L}(x, y, t)$ for any $\hat{y}_i^t \in \mathcal{Y}$. If we consider a valid case where each label in $\mathcal{G}_t \cup \mathcal{Y}_I \setminus \{y\}$ is exactly sampled to constitute $\hat{\mathbf{y}}_{N-1}^{(t)}$ when calculating $\hat{L}(x, y, t)$ in Eq. (39), we have $N = K + L/T$ such that $N \to +\infty$ requires $L/T \to +\infty$, which results in rewriting Eq. (39) as follows:

$$
\lim_{N \to +\infty} E_{\boldsymbol{\theta}}(\mathbf{x}) = \lim_{\substack{T \to +\infty \\ N \to +\infty}} \log \sum_{y \in \mathcal{Y}_{\mathrm{I}}} \frac{N}{T} \sum_{i=1}^{T} \hat{L}(x, y, t)
$$

$$
= \lim_{\substack{T \to +\infty \\ N \to +\infty}} \log \sum_{y \in \mathcal{Y}_{\mathrm{I}}} \frac{K + L/T}{T} \sum_{i=1}^{T} \frac{\exp h_{\boldsymbol{\theta}}(\mathbf{x}, y)}{\sum_{\hat{y}_j \in \mathcal{G}_t \cup \mathcal{Y}_I} \exp h_{\boldsymbol{\theta}}(\mathbf{x}, \hat{y}_j)} \tag{41}
$$

$$
= \lim_{\substack{T \to +\infty \\ L/T \to +\infty}} \log(K + L/T) S_{\mathrm{NegLabel}}(\mathbf{x}; \boldsymbol{\theta}).
$$

Step 2 and Step 3 implies this result. $\qquad \square$

**Remarks.** We note that the same ideas of the derivation in Step 3 have been witnessed in contrastive learning that is known for minimizing InfoNCE [69, 60] of the form $-\mathbb{E}_{(x,y) \in \mathbb{P}_{XY}} R(x, y)$ where

$$
R(x, y) = \mathbb{E}_{\hat{\mathbf{y}}_N \in \mathbb{P}_Y^N} \left[ \log \frac{\exp h_\theta(x, y)}{\exp h_\theta(x, y) + \sum_{j=1}^{N} \exp h_\theta(x, \hat{y}_j)} \right]. \tag{42}
$$

Recalling the batch-wise empirical loss of contrastive learners such as SimCLR [5] ($x$ and $y$ share a same modality) and CLIP ($x$ and $y$ are with different modalities), i.e.,

$$
-\frac{1}{B} \sum_{i=1}^{B} \log \frac{\exp h_\theta(x_i, y_i)}{\exp h_\theta(x_i, y_i) + \sum_{j \neq i} \exp h_\theta(x_i, y_j)},
$$

where $\mathcal{B} = \{(x_i, y_i)\}_{i=1}^{B}$ is the current training batch, one can check that each given $R(x, y)$ is estimated by exactly sampling $\hat{\mathbf{y}}_N$ as $\{y_j | (x_j, y_j) \in \mathcal{B} \text{ and } y_j \neq y\}$.

# E  Proof of Theorem 3

**Definition 2** (**Mutual Information (MI)**). Given two random variables $X$ and $Y$, the MI between $X$ and $Y$ is the Kullback-Leibler (KL) divergence between the joint distribution $\mathbb{P}_{XY}$ and the product of marginal distributions $\mathbb{P}_X \mathbb{P}_Y$, i.e.,

$$I(X;Y) \triangleq D_{KL}(\mathbb{P}_{XY} || \mathbb{P}_X \mathbb{P}_Y) = \mathbb{E}_{(\mathbf{x},y) \sim \mathbb{P}_{XY}} \left[ \log \frac{p_{XY}(\mathbf{x},y)}{p_X(\mathbf{x}) p_Y(y)} \right] = \mathbb{E}_{(\mathbf{x},y) \sim \mathbb{P}_{XY}} \left[ \mathcal{W}(\mathbf{x},y) \right]. \tag{43}$$

where $\mathcal{W}(\mathbf{x},y)$ is the PMI defined in the main paper.

**Lemma 1.** *Given three random variables $X$, $Y$, and $\tilde{X}$, the mutual information $I(X;Y|V)$ can be decomposed into the following two ways:*

$$I(X, \tilde{X}; Y) = I(X;Y) + I(\tilde{X};Y|X) = I(\tilde{X};Y) + I(X;Y|\tilde{X}), \tag{44}$$

*where $I(X;Y|\tilde{X})$, i.e., the MI between $X$ and $Y$ conditioned on $\tilde{X}$, is defined as follows:*

$$I(X;Y|\tilde{X}) \triangleq \mathbb{E}_{(\mathbf{x},y,\tilde{\mathbf{x}}) \sim \mathbb{P}_{XY\tilde{X}}} \left[ \log \frac{p_{XY|\tilde{X}}(\mathbf{x},y|\tilde{\mathbf{x}})}{p_{X|\tilde{X}}(\mathbf{x}|\tilde{\mathbf{x}}) p_{Y|\tilde{X}}(y|\tilde{\mathbf{x}})} \right] = \mathbb{E}_{(\mathbf{x},y,\tilde{\mathbf{x}}) \sim \mathbb{P}_{XY\tilde{X}}} \left[ \mathcal{W}(\mathbf{x},y|\tilde{\mathbf{x}}) \right]. \tag{45}$$

As a reminder, Theorem 3 is stated again as follows:

**Theorem 3.** *For any $\mathbf{x} \in \mathcal{X}$ and $y \in \mathcal{Y}$, let $\tilde{\mathbf{x}} = \mathcal{T}(\mathbf{x})$ be a sub-view of the input $\mathbf{x}$, $\mathcal{W}(\mathbf{x},y)$ can be decomposed into the following two terms:*

$$\mathcal{W}(\mathbf{x},y) = \mathcal{W}(\tilde{\mathbf{x}},y) + \mathcal{W}(\mathbf{x},y|\tilde{\mathbf{x}}), \tag{46}$$

*where $\mathcal{W}(\mathbf{x},y|\tilde{\mathbf{x}})$, i.e., the PMI between $\mathbf{x}$ and $y$ conditioned on $\tilde{\mathbf{x}}$, is defined as follows:*

$$\begin{aligned}
\mathcal{W}(\mathbf{x},y|\tilde{\mathbf{x}}) &\triangleq \log \frac{p_{XY|\tilde{X}}(\mathbf{x},y|\tilde{\mathbf{x}})}{p_{X|\tilde{X}}(\mathbf{x}|\tilde{\mathbf{x}}) p_{Y|\tilde{X}}(y|\tilde{\mathbf{x}})} \\
&= \log \frac{p_{Y|X\tilde{X}}(y|\mathbf{x},\tilde{\mathbf{x}})}{p_{Y|\tilde{X}}(y|\tilde{\mathbf{x}})}.
\end{aligned} \tag{47}$$

*Proof.* For any $\mathbf{x} \in \mathcal{X}$ and $y \in \mathcal{Y}$, let $\tilde{\mathbf{x}} = \mathcal{T}(\mathbf{x})$ be a sub-view of the input $\mathbf{x}$, the data processing inequality implies that $I(\tilde{\mathbf{x}};y|\mathbf{x}) = 0$, which means that $I(X;Y) = I(\tilde{X};Y) + I(X;Y|\tilde{X})$.

Given that

$$\begin{aligned}
\mathbb{E}_{(\tilde{\mathbf{x}},y) \sim \mathbb{P}_{\tilde{X}Y}} \left[ \log \frac{p_{\tilde{X}Y}(\tilde{\mathbf{x}},y)}{p_{\tilde{X}}(\tilde{\mathbf{x}}) p_Y(y)} \right] &= \sum_{\tilde{\mathbf{x}}} \sum_y p_{\tilde{X}Y}(\tilde{\mathbf{x}},y) \left[ \log \frac{p_{\tilde{X}Y}(\tilde{\mathbf{x}},y)}{p_{\tilde{X}}(\tilde{\mathbf{x}}) p_Y(y)} \right] \\
&= \sum_{\tilde{\mathbf{x}}} \sum_y \left( \sum_{\mathbf{x}} p_{XY\tilde{X}}(\mathbf{x},y,\tilde{\mathbf{x}}) \right) \left[ \log \frac{p_{\tilde{X}Y}(\tilde{\mathbf{x}},y)}{p_{\tilde{X}}(\tilde{\mathbf{x}}) p_Y(y)} \right] \\
&= \sum_{\tilde{\mathbf{x}}} \sum_y \sum_{\mathbf{x}} p_{XY\tilde{X}}(\mathbf{x},y,\tilde{\mathbf{x}}) \left[ \log \frac{p_{\tilde{X}Y}(\tilde{\mathbf{x}},y)}{p_{\tilde{X}}(\tilde{\mathbf{x}}) p_Y(y)} \right] \\
&= \mathbb{E}_{(\mathbf{x},y,\tilde{\mathbf{x}}) \sim \mathbb{P}_{XY\tilde{X}}} \left[ \log \frac{p_{\tilde{X}Y}(\tilde{\mathbf{x}},y)}{p_{\tilde{X}}(\tilde{\mathbf{x}}) p_Y(y)} \right]
\end{aligned} \tag{48}$$

and

$$\begin{aligned}
\mathbb{E}_{(\mathbf{x},y) \sim \mathbb{P}_{XY}} \left[ \log \frac{p_{XY}(\mathbf{x},y)}{p_X(\mathbf{x}) p_Y(y)} \right] &= \sum_{\mathbf{x}} \sum_y p_{XY}(\mathbf{x},y) \left[ \log \frac{p_{XY}(\mathbf{x},y)}{p_X(\mathbf{x}) p_Y(y)} \right] \\
&= \sum_{\tilde{\mathbf{x}}} \sum_y \left( \sum_{\mathbf{x}} p_{XY\tilde{X}}(\mathbf{x},y,\tilde{\mathbf{x}}) \right) \left[ \log \frac{p_{XY}(\mathbf{x},y)}{p_X(\mathbf{x}) p_Y(y)} \right] \\
&= \sum_{\mathbf{x}} \sum_y \sum_{\tilde{\mathbf{x}}} p_{XY\tilde{X}}(\mathbf{x},y,\tilde{\mathbf{x}}) \left[ \log \frac{p_{XY}(\mathbf{x},y)}{p_X(\mathbf{x}) p_Y(y)} \right] \\
&= \mathbb{E}_{(\mathbf{x},y,\tilde{\mathbf{x}}) \sim \mathbb{P}_{XY\tilde{X}}} \left[ \log \frac{p_{XY}(\mathbf{x},y)}{p_X(\mathbf{x}) p_Y(y)} \right],
\end{aligned} \tag{49}$$

we have the following:

$$I(X;Y) = I(\tilde{X};Y) + I(X;Y|\tilde{X})$$

$$\Leftrightarrow \mathbb{E}_{(\mathbf{x},y)\sim\mathbb{P}_{XY}} \left[\log \frac{p_{XY}(\mathbf{x},y)}{p_X(\mathbf{x})p_Y(y)}\right] = \mathbb{E}_{(\mathbf{x},y)\sim\mathbb{P}_{\tilde{X}Y}} \left[\log \frac{p_{\tilde{X}Y}(\tilde{\mathbf{x}},y)}{p_{\tilde{X}}(\tilde{\mathbf{x}})p_Y(y)}\right] + \mathbb{E}_{(\mathbf{x},y,\tilde{\mathbf{x}})\sim\mathbb{P}_{XY\tilde{X}}} \left[\log \frac{p_{XY|\tilde{X}}(\mathbf{x},y|\tilde{\mathbf{x}})}{p_{X|\tilde{X}}(\mathbf{x}|\tilde{\mathbf{x}})p_{Y|\tilde{X}}(y|\tilde{\mathbf{x}})}\right]$$

$$\Leftrightarrow \mathbb{E}_{(\mathbf{x},y,\tilde{\mathbf{x}})\sim\mathbb{P}_{XY\tilde{X}}} \left[\log \frac{p_{XY}(\mathbf{x},y)}{p_X(\mathbf{x})p_Y(y)}\right] = \mathbb{E}_{(\mathbf{x},y,\tilde{\mathbf{x}})\sim\mathbb{P}_{XY\tilde{X}}} \left[\log \frac{p_{\tilde{X}Y}(\tilde{\mathbf{x}},y)}{p_{\tilde{X}}(\tilde{\mathbf{x}})p_Y(y)}\right] + \mathbb{E}_{(\mathbf{x},y,\tilde{\mathbf{x}})\sim\mathbb{P}_{XY\tilde{X}}} \left[\log \frac{p_{XY|\tilde{X}}(\mathbf{x},y|\tilde{\mathbf{x}})}{p_{X|\tilde{X}}(\mathbf{x}|\tilde{\mathbf{x}})p_{Y|\tilde{X}}(y|\tilde{\mathbf{x}})}\right]$$

$$\Leftrightarrow \mathbb{E}_{(\mathbf{x},y,\tilde{\mathbf{x}})\sim\mathbb{P}_{XY\tilde{X}}} \left[\log \frac{p_{XY}(\mathbf{x},y)}{p_X(\mathbf{x})p_Y(y)} - \log \frac{p_{\tilde{X}Y}(\tilde{\mathbf{x}},y)}{p_{\tilde{X}}(\tilde{\mathbf{x}})p_Y(y)} - \log \frac{p_{XY|\tilde{X}}(\mathbf{x},y|\tilde{\mathbf{x}})}{p_{X|\tilde{X}}(\mathbf{x}|\tilde{\mathbf{x}})p_{Y|\tilde{X}}(y|\tilde{\mathbf{x}})}\right] = 0$$

$$\Leftrightarrow \mathbb{E}_{(\mathbf{x},y,\tilde{\mathbf{x}})\sim\mathbb{P}_{XY\tilde{X}}} \left[\mathcal{W}(\mathbf{x},y) - \mathcal{W}(\tilde{\mathbf{x}},y) - \mathcal{W}(\mathbf{x},y|\tilde{\mathbf{x}})\right] = 0$$

$$\Leftrightarrow \mathcal{W}(\mathbf{x},y) = \mathcal{W}(\tilde{\mathbf{x}},y) + \mathcal{W}(\mathbf{x},y|\tilde{\mathbf{x}})$$

(50)

$\square$

# F    Proof of Theorem 4

As a reminder, Theorem 4 is stated as follows:

**Theorem 4.** *Let us define $h_{\boldsymbol{\theta}}(\mathbf{x},\tilde{\mathbf{x}},y) \triangleq \mathbf{r}^\top[\beta\tilde{\mathbf{z}} + (1-\beta)\mathbf{z}]/\kappa$ and $h_{\boldsymbol{\theta}}(\tilde{\mathbf{x}},y) = \tilde{\mathbf{z}}^\top\mathbf{r}/\tau$. If we, following prior works [56, 38] respectively, assume that $h_{\boldsymbol{\theta}}(\tilde{\mathbf{x}},y) = \log \frac{p_{Y|\tilde{X}}(y|\tilde{\mathbf{x}})}{p_Y(y)} + c(\tilde{\mathbf{x}})$ with $c(\tilde{\mathbf{x}})$ as a constant term depending on $\tilde{\mathbf{x}}$, and that $h_{\boldsymbol{\theta}}(\mathbf{x},\tilde{\mathbf{x}},y) = \log \frac{p_{Y|X\tilde{X}}(y|\mathbf{x},\tilde{\mathbf{x}})}{p_{Y|\tilde{X}}(y|\tilde{\mathbf{x}})} + c(\mathbf{x},\tilde{\mathbf{x}})$ with $c(\mathbf{x},\tilde{\mathbf{x}})$ as a constant term depending on $\mathbf{x}$ and $\tilde{\mathbf{x}}$, we then have the following for $\alpha = 1$:*

$$E(\mathbf{x}) = \lim_{N\to+\infty} E_{\boldsymbol{\theta}}(\mathbf{x}) = \lim_{\substack{T\to+\infty \\ L/T\to+\infty}} \log(K+L)S_{ours}(\mathbf{x};\boldsymbol{\theta}). \tag{51}$$

*Proof.* **Step 1:** Given that $\alpha = 1$, we have

$$E(\mathbf{x}) = \frac{1}{\alpha}\log\sum_{y\in\mathcal{Y}_\mathrm{I}}\exp\left[\alpha\mathcal{W}(x,y)\right] = \log\sum_{y\in\mathcal{Y}_\mathrm{I}}\exp\left[\log\frac{p_{Y|X}(y|\mathbf{x})}{p_Y(y)}\right] = \log\sum_{y\in\mathcal{Y}_\mathrm{I}}\frac{p_{Y|X}(y|\mathbf{x})}{p_Y(y)}. \tag{52}$$

Given that $h_{\boldsymbol{\theta}}(\tilde{\mathbf{x}},y) = \log\frac{p_{Y|\tilde{X}}(y|\tilde{\mathbf{x}})}{p_Y(y)} + c(\tilde{\mathbf{x}})$, we have

$$\begin{aligned}
\lim_{N\to+\infty}\hat{\mathcal{W}}(\tilde{\mathbf{x}},y;\boldsymbol{\theta}) &= \lim_{N\to+\infty}\log\frac{\hat{p}_{Y|\tilde{X}}(y|\tilde{\mathbf{x}};\boldsymbol{\theta})}{p_Y(y)} \\
&= \lim_{N\to+\infty}\log\mathbb{E}_{\hat{\mathbf{y}}_{N-1}\sim\mathbb{P}_Y^{N-1}}\left[\frac{N\cdot\exp h_{\boldsymbol{\theta}}(\tilde{\mathbf{x}},y)}{\exp h_{\boldsymbol{\theta}}(\tilde{\mathbf{x}},y) + \sum_{j=1}^{N-1}\exp h_{\boldsymbol{\theta}}(\tilde{\mathbf{x}},\hat{y}_j)}\right] \\
&= \log\frac{p_{Y|\tilde{X}}(y|\tilde{\mathbf{x}})}{p_Y(y)} = \mathcal{W}(\tilde{\mathbf{x}},y)
\end{aligned} \tag{53}$$

**Step 2:**

Since $h_{\boldsymbol{\theta}}(\mathbf{x}, \tilde{\mathbf{x}}, y) = \log \frac{p_{Y|X\tilde{X}}(y|\mathbf{x},\tilde{\mathbf{x}})}{p_{Y|\tilde{X}}(y|\tilde{\mathbf{x}})} + c(\mathbf{x}, \tilde{\mathbf{x}})$, by closely following Eq. (37) and Eq. (38), Eq. (53) implies that

$$
\begin{aligned}
\lim_{N\to+\infty} E_{\boldsymbol{\theta}}(\mathbf{x}) &= \lim_{N\to+\infty} \log \sum_{y\in\mathcal{Y}_{\mathrm{I}}} \frac{\exp\left[\hat{\mathcal{W}}(\tilde{\mathbf{x}}, y; \boldsymbol{\theta}) + h_{\boldsymbol{\theta}}(\mathbf{x}, \tilde{\mathbf{x}}, y)\right]}{\mathbb{E}_{\hat{y}\sim\mathbb{P}_Y}\left[\exp\left[\hat{\mathcal{W}}(\tilde{\mathbf{x}}, \hat{y}; \boldsymbol{\theta}) + h_{\boldsymbol{\theta}}(\mathbf{x}, \tilde{\mathbf{x}}, \hat{y})\right]\right]} \\
&= \log \sum_{y\in\mathcal{Y}_{\mathrm{I}}} \frac{\exp\left[\mathcal{W}(\tilde{\mathbf{x}}, y) + h_{\boldsymbol{\theta}}(\mathbf{x}, \tilde{\mathbf{x}}, y)\right]}{\mathbb{E}_{\hat{y}\sim\mathbb{P}_Y}\left[\exp\left[\mathcal{W}(\tilde{\mathbf{x}}, \hat{y}) + h_{\boldsymbol{\theta}}(\mathbf{x}, \tilde{\mathbf{x}}, \hat{y})\right]\right]} \\
&= \log \sum_{y\in\mathcal{Y}_{\mathrm{I}}} \frac{\exp\left[\log\frac{p_{Y|\tilde{X}}(y|\tilde{\mathbf{x}})}{p_Y(y)} + \log\frac{p_{Y|X\tilde{X}}(y|\mathbf{x},\tilde{\mathbf{x}})}{p_{Y|\tilde{X}}(y|\tilde{\mathbf{x}})} + c(\mathbf{x}, \tilde{\mathbf{x}})\right]}{\mathbb{E}_{\hat{y}\sim\mathbb{P}_Y}\left[\exp\left[\log\frac{p_{Y|\tilde{X}}(\hat{y}|\tilde{\mathbf{x}})}{p_Y(\hat{y})} + \log\frac{p_{Y|X\tilde{X}}(\hat{y}|\mathbf{x},\tilde{\mathbf{x}})}{p_{Y|\tilde{X}}(\hat{y}|\tilde{\mathbf{x}})} + c(\mathbf{x}, \tilde{\mathbf{x}})\right]\right]} \\
&= \log \sum_{y\in\mathcal{Y}_{\mathrm{I}}} \frac{\frac{p_{Y|X\tilde{X}}(y|\mathbf{x},\tilde{\mathbf{x}})}{p_Y(y)}}{\mathbb{E}_{\hat{y}\sim\mathbb{P}_Y}\left[\frac{p_{Y|X\tilde{X}}(\hat{y}|\mathbf{x},\tilde{\mathbf{x}})}{p_Y(\hat{y})}\right]} \\
&= \log \sum_{y\in\mathcal{Y}_{\mathrm{I}}} \frac{p_{Y|X\tilde{X}}(y|\mathbf{x},\tilde{\mathbf{x}})}{p_Y(y)} = E(\mathbf{x})
\end{aligned}
\tag{54}
$$

**Step 3:**

Let $\hat{y}_i^{(t)}$ be the $i$-th sample drawn at the $t$-th round, the Law of Large Numbers implies

$$
\begin{aligned}
\mathcal{W}(\tilde{\mathbf{x}}, y) &= \lim_{N\to+\infty} \log \mathbb{E}_{\hat{\mathbf{y}}_{N-1}\sim\mathbb{P}_Y^{N-1}} \left[\frac{N \cdot \exp h_{\boldsymbol{\theta}}(\tilde{\mathbf{x}}, y)}{\exp h_{\boldsymbol{\theta}}(\tilde{\mathbf{x}}, y) + \sum_{j=1}^{N-1} \exp h_{\boldsymbol{\theta}}(\tilde{\mathbf{x}}, \hat{y}_j)}\right] \\
&= \lim_{N\to+\infty} \lim_{T\to+\infty} \frac{1}{T} \sum_{i=1}^{T} \frac{N \cdot \exp h_{\boldsymbol{\theta}}(\mathbf{x}, y)}{\exp h_{\boldsymbol{\theta}}(\mathbf{x}, y) + \sum_{j=1}^{N-1} \exp h_{\boldsymbol{\theta}}(\mathbf{x}, \hat{y}_j^{(t)})}.
\end{aligned}
\tag{55}
$$

Similarly, by taking $\mathcal{G}_t \cup \mathcal{Y}_{\mathrm{I}} \setminus \{y\}$ as the examplar of $\left\{\hat{y}_i^{(t)}\right\}_{i=1}^{N-1}$ for $\mathcal{W}(\tilde{\mathbf{x}}, y)$ in Eq. (55), we have $N = K + L/T$ such that $N \to +\infty$ requires $L/T \to +\infty$, which results in rewriting Eq. (55) as follows:

$$
\begin{aligned}
\mathcal{W}(\tilde{\mathbf{x}}, y) &= \lim_{\substack{T\to+\infty \\ L/T\to+\infty}} \log \sum_{y\in\mathcal{Y}_{\mathrm{I}}} \frac{1}{T} \sum_{t=1}^{T} \frac{\exp(\tilde{\mathbf{z}}^\top \mathbf{r}/\tau)}{\sum_{y_j\in\mathcal{G}_t\cup\mathcal{Y}_{\mathrm{I}}} \exp(\tilde{\mathbf{z}}^\top \mathbf{r}_j/\tau)} + \log(K + L/T) \\
&= \lim_{\substack{T\to+\infty \\ L/T\to+\infty}} \Lambda(\tilde{\mathbf{x}}, y)
\end{aligned}
\tag{56}
$$

Given that $N = T \cdot L/T \to +\infty$ when $L/T \to +\infty$ and $T \to +\infty$, as implied by the Law of Large Number, combining Eq. (54) with Eq. (56) arrives at the following:

$$
\begin{aligned}
&\lim_{\substack{T\to+\infty \\ L/T\to+\infty}} \log(K + L) S_{\mathrm{ours}}(\mathbf{x}; \boldsymbol{\theta}) \\
&= \lim_{\substack{T\to+\infty \\ L/T\to+\infty}} \log \sum_{y\in\mathcal{Y}_{\mathrm{I}}} \frac{\exp\left[\Lambda(\tilde{\mathbf{x}}, y; \boldsymbol{\theta}) + h_{\boldsymbol{\theta}}(\mathbf{x}, \tilde{\mathbf{x}}, y)\right]}{\sum_{j=1}^{K+L} \exp\left[\Lambda(\tilde{\mathbf{x}}, y_j; \boldsymbol{\theta}) + h_{\boldsymbol{\theta}}(\mathbf{x}, \tilde{\mathbf{x}}, y_j)\right]} + \log(K + L) \\
&= \lim_{L\to+\infty} \log \sum_{y\in\mathcal{Y}_{\mathrm{I}}} \frac{\exp\left[\mathcal{W}(\tilde{\mathbf{x}}, y) + h_{\boldsymbol{\theta}}(\mathbf{x}, \tilde{\mathbf{x}}, y)\right]}{\sum_{j=1}^{K+L} \exp\left[\mathcal{W}(\tilde{\mathbf{x}}, y) + h_{\boldsymbol{\theta}}(\mathbf{x}, \tilde{\mathbf{x}}, y_j)\right]} + \log(K + L) \\
&= \log \sum_{y\in\mathcal{Y}_{\mathrm{I}}} \frac{\exp\left[\mathcal{W}(\tilde{\mathbf{x}}, y) + h_{\boldsymbol{\theta}}(\mathbf{x}, \tilde{\mathbf{x}}, y)\right]}{\mathbb{E}_{\hat{y}\sim\mathbb{P}_Y}\left[\exp\left[\mathcal{W}(\tilde{\mathbf{x}}, \hat{y}) + h_{\boldsymbol{\theta}}(\mathbf{x}, \tilde{\mathbf{x}}, \hat{y})\right]\right]} = E(\mathbf{x})
\end{aligned}
\tag{57}
$$

Step 2 and Step 3 imply this result. $\qquad\square$

Table 4: Comparison with different VLM architectures on ImageNet-1K (ID).All values are percentages. ↑ indicates larger values are better and vice versa. The best results in the last two columns are shown in bold per ID dataset. Results are averaged over 5 independent runs.

| ID Dataset | Method | iNaturalist | | SUN | | Places | | Textures | | Average | |
|---|---|---|---|---|---|---|---|---|---|---|---|
| | | AUROC↑ | FPR95↓ | AUROC↑ | FPR95↓ | AUROC↑ | FPR95↓ | AUROC↑ | FPR95↓ | AUROC↑ | FPR95↓ |
| GroupViT [71] | NegLabel | 98.07 | 8.60 | 91.52 | 35.12 | 88.85 | 41.63 | 88.45 | 47.06 | 91.72 | 33.10 |
| | Ours | 99.52 | 6.86 | 95.52 | 25.08 | 92.24 | 37.44 | 91.28 | 46.15 | **94.70** | **29.08** |

# G  Additional Experiments

## G.1  More Backbones

## G.2  Cropping vs Cutout

Table 5: Ablation study on ImageNet-1K w.r.t the choice of obtaining subviews. ↑ indicates larger values are better and vice versa. Results are averaged over 5 independent runs.

| Method | iNaturalist | | SUN | | Places | | Textures | | Average | |
|---|---|---|---|---|---|---|---|---|---|---|
| | AUROC↑ | FPR95↓ | AUROC↑ | FPR95↓ | AUROC↑ | FPR95↓ | AUROC↑ | FPR95↓ | AUROC↑ | FPR95↓ |
| Cropping | 99.64 | 1.04 | 96.32 | 18.45 | 95.81 | 31.15 | 92.15 | 38.79 | 96.00 | 22.36 |
| Cutout | 99.52 | 1.26 | 95.08 | 19.08 | 95.46 | 32.44 | 91.50 | 39.38 | 95.39 | 23.04 |

# H  Limitations

This paper directly uses the off-the-shelf negative labels mined by NegLabel for PMI estimation. It will be exciting to what makes good negative labels for OOD detection with pre-trained VLMs.

# I  Broader impacts

Our project aims to improve the reliability and safety of modern machine learning models, which leads to benefits and societal impacts, particularly for safety-critical applications such as autonomous driving. Our study does not involve any human subjects or violation of legal compliance. We do not anticipate any potentially harmful consequences to our work.

# J  Stability

To verify that our method consistently provides strong performance, we run with 10 independent seeds for ImageNet-1K and report the average and standard deviation of FPR95 and AUROC as follows.

Table 6: Ablation on stability. OOD detection performance based on CLIP-B/16 of our method on ImageNet-1K. Results are averaged over 5 independent runs.

| iNaturalist | | SUN | | Places | | Textures | |
|---|---|---|---|---|---|---|---|
| FPR95↓ | AUROC↑ | FPR95↓ | AUROC↑ | FPR95↓ | AUROC↑ | FPR95↓ | AUROC↑ |
| 0.0 | 0.1 | 1.2 | 0.5 | 3.0 | 1.0 | 1.6 | 0.6 |

