# OpenReview forum: "An Information-theoretical Framework for Understanding Out-of-distribution Detection with Pretrained Vision-Language Models"
_NeurIPS.cc/2025/Conference — NeurIPS 2025 poster_

### Official Review · Reviewer_jjLB · 2025-07-01

**Clarity:** 3
**Significance:** 2
**Originality:** 2
**Rating:** 4
**Confidence:** 4

**Summary:**

This paper introduces an information theory-based approach to enhance OOD detection using pretrained vision-language models such as CLIP. The method involves stochastic estimation of pointwise mutual information (PMI) and its decomposition via the chain rule. The authors validate the effectiveness of the proposed method through experiments on several classification benchmarks.

**Questions:**

What is the underlying intuition of the proposed method? Are there any illustrations or pseudocode to help convey the core idea?
It is important to link the evaluation results to the underlying effects of the proposed method. Rather than simply showing that it outperforms existing approaches on standard benchmarks, I’m more interested in understanding why and how it improves OOD detection

**Ethical Concerns:**

["NO or VERY MINOR ethics concerns only"]

**Final Justification:**

Thank you to the authors for providing further explanations and conducting additional experiments that effectively addressed my main concerns. Based on these clarifications and new results, I have decided to raise my score. I hope that these insights and the outcomes of the new experiments will be incorporated into the final version of the paper, should it be accepted.

**Limitations:**

yes

**Paper Formatting Concerns:**

No.

**Quality:**

2

**Strengths And Weaknesses:**

Strengths:

Shows promising results on standard OOD detection tasks

Weakness:

1. The intuition behind the proposed method is not clearly presented, and the absence of illustrations or pseudocode makes the results difficult to reproduce
2. The evaluations do not clearly address why and how the proposed method enhances OOD detection
3. The evaluations are primarily conducted on standard classification benchmarks to demonstrate the effectiveness of the proposed method; including benchmarks from real-world applications would further strengthen the results

---

> ### Author Rebuttal · Authors · 2025-07-30
>
> We thank Reviewer jjLB for valuable comments. As to the questions and suggestions you raise, we took them seriously. Our response is as follows.
>
> ## 1. What is the underlying intuition of the proposed method and why and how the proposed method enhances OOD detection
>
> The core intuition of our method starts from the thorectical analysis that representative post-hoc CLIP-based OOD detectors (MCM and NegLabel) can be understand as stochastic estimators of Pointwise Mutual Information (PMI) estimator. We, motivated by thedivide-and-conquer philosophy, propose to decompose PMI into two subterms, i.e., $\mathcal{W}(\mathbf{x},y)=\mathcal{W}(\tilde{\mathbf{x}},y)+\mathcal{W}(\mathbf{x},y|\tilde{\mathbf{x}})$ so as to transfer the estimation of $\mathcal{W}(\mathbf{x},y)$ into separately estimating $\mathcal{W}(\tilde{\mathbf{x}},y)$ and $\mathcal{W}(\mathbf{x},y|\tilde{\mathbf{x}})$.
>
> The theoretical benefits of our PMI decomposition to OOD detection can be summarized in two aspects:
>
> - Our PMI decompositon reduces the complexity of estimating $\mathcal{W}(\mathbf{x},y)$. This is becuase
>
> a) estimating $\mathcal{W}(\tilde{\mathbf{x}},y)$ involves estimating $p(\tilde{\mathbf{x}},y)$, which is simpler than estimating $p({\mathbf{x}},y)$ as $\tilde{\mathbf{x}}$ is a subview of $\mathbf{x}$ to make the space of possible $\tilde{\mathbf{x}}$ and $y$ lower-dimensional.
>
> b) estimating $\mathcal{W}(\mathbf{x},y|\tilde{\mathbf{x}})$ involves estimating $p(\mathbf{x},y|\tilde{\mathbf{x}})$, which is simpler than estimating $p({\mathbf{x}},y)$ as $\tilde{\mathbf{x}}$ is a subview of $\mathbf{x}$ to make the conditioned space of possible $\mathbf{x}$ and $y$ lower-dimensional.
>
> - We show in Theorem 4 that our PMI decompositon provably increases the upper bound of the the estimated PMI to $\log(K+L)$ from NegLabel’s $\log(K + L/T)$ or MCM's $\log K$, therefore reducing the underestimation bias.
>
> ## 2. Are there any illustrations or pseudocode to help convey the core idea?
>
> The pseudo algorithm of our method is given as follows:
> >**Input**: input image $\mathbf{x}$, ID labels $\mathcal{Y}\_{I}$, Negative labels $\mathcal{Y}\_{N}$, pre-trained CLIP model parameters $\theta$, sub-view generator $\mathcal{T}$, pre-defined critics $h\_{\boldsymbol{\theta}}(\mathbf{x},\hat{y})$ and $h\_{\boldsymbol{\theta}}(\mathbf{x},\tilde{\mathbf{x}},\hat{y})$
> >
> >**Output:** Our proposed OOD scoring function $S\_{ours}(\mathbf{x})$
>
> >Generate a subview via $\tilde{\mathbf{x}}=\mathcal{T}(\mathbf{x})$
> >
> >Randomly divide $\mathcal{Y}\_{N}$ into $T$ non-overlapping groups, i.e., $\mathcal{Y}\_{N}=\cup_{i=1}^L \mathcal{G}_i$
> >
> > **For** each $y$ in $\mathcal{Y}\_{I}$ **do**
> >> $\hat{\mathcal{W}}(\tilde{\mathbf{x}},y;\boldsymbol{\theta})\leftarrow\log \frac{1}{T}\sum\_{t=1}^T\frac{ \exp h_{\boldsymbol{\theta}}(\mathbf{x},y)}{\underset{{y\_j\in\mathcal{G}\_t\cup\mathcal{Y}\_I}}{\sum}\exp h_{\boldsymbol{\theta}}(\mathbf{x},y_j)}+\log(K+L/T)$
> >
> > **end For**
> >
> >$S\_{ours}(\mathbf{x})\leftarrow\sum\_{y\in\mathcal{Y}\_{I}}\frac{\exp \big [\hat{\mathcal{W}}(\tilde{\mathbf{x}},y;\boldsymbol{\theta})+h\_{\boldsymbol{\theta}}(\mathbf{x},\tilde{\mathbf{x}},y)\big ]}{\underset{{y\_j\in\mathcal{G}\_t\cup\mathcal{Y}\_I}}{\sum} \exp \big [\hat{\mathcal{W}}(\tilde{\mathbf{x}},\hat{y};\boldsymbol{\theta})+h\_{\boldsymbol{\theta}}(\mathbf{x},\tilde{\mathbf{x}},\hat{y})\big ]}$
>
> ## 3. Including real-world benchmarks would further strengthen the results.
>
> Thanks for your constructive. As per your advice, we conduct experiments on the BIMCV-COVID19+ dataset [a], following the same setup as AdaNeg [b]. Specifically, we select BIMCV as the ID dataset, which includes chest X-ray images CXR (CR, DX) of COVID-19 patients and healthy individuals. For the OOD datasets, we, following AdaNeg [b], use CT-SCAN and X-Ray-Bone datasets. The CT-SCAN dataset includes computed tomography (CT) images of COVID-19 patients and healthy individuals, while the X-Ray-Bone dataset contains X-ray images of hands. We report AUROC as follows:
> |AUROC|CT-SCAN|X-Ray-Bone|AVG|
> |-|-|-|-|
> |ours|74.63|99.82|87.26|
> |NegLabel|63.53|99.68|81.61|
> |ours+AdaNeg|95.16|99.99|97.58|
> |NegLabel+AdaNeg|93.48|99.99|96.74|
>
> [a] Bimcv covid-19+: a large annotated dataset of rx and ct images from covid-19 patients
>
> [b] AdaNeg: Adaptive Negative Proxy Guided OODDetection with Vision-Language Models

---

> > ### Author Response · Authors · 2025-08-06
> >
> > Dear Reviewer jjLB:
> >
> > Thanks again for your detailed and insightful ​​reviews​​! We would like to double-check and see if our responses have addressed your concerns, as the deadline for the rebuttal period is around the corner. If you have any further questions, please feel free to let us know. We are more than happy to clarify more about our paper and discuss it further with you.
> >
> > Thanks.

---

> ### Author Response · Authors · 2025-08-08
>
> Thank you so much for your recognition of our work and your constructive comment, which is very helpful in improving the quality of our work. We will incorportate these insights and the outcomes of the new experiments into the revised version of the paper.

---

### Official Review · Reviewer_oce3 · 2025-07-01

**Clarity:** 3
**Significance:** 2
**Originality:** 3
**Rating:** 4
**Confidence:** 3

**Summary:**

This paper proposed an information theory-based method to solve the OOD detection via the VLM. The main motivation for this paper is that point-wise mutual information (PMI) could clearly differentiate the (in-distribution) ID and OOD samples. The problem here is how to calculate the PMI. This paper proposes an approximation method to estimate the PMI. The experimental results show that PMI could achieve the better performance on OOD detection.

**Questions:**

No other question, please refer Weaknesses.

**Ethical Concerns:**

["NO or VERY MINOR ethics concerns only"]

**Final Justification:**

This paper establishes a connection between information and CLIP in OOD detection through PMI. The initial concerns include unclear contribution and a lack of the necessary ablation study. After rebuttal, the author addresses these concerns. However, the problem is that the proposed PMI belongs to the theory framework of energy-based functions, thereby weakening its contribution to be significant. Thus, I think borderline accept is suitable for this paper.

**Limitations:**

yes

**Quality:**

2

**Strengths And Weaknesses:**

Strengths:
1. The writing for this paper is good.

2. The proposed theory is solid.


Weaknesses:

1. Limited novelty. The motivation of the paper is to compute the OOD score using an energy function. However, this idea has already been proposed in "Energy-based Out-of-Distribution Detection." The authors suggest that VLMs (Visual-Language Models) can estimate PMI to derive the energy function, but this innovation seems insufficient, as the method does not fundamentally differ from existing energy-based methods. The authors have not clearly distinguished their approach from the one presented in "Energy-based Out-of-Distribution Detection," which weakens the novelty of the proposed work.

2. Lacking an ablation study about the computation cost. The proposed method requires sampling multiple sub-views for Monte Carlo estimation. However, there is no discussion on the computational overhead incurred by this process. It would be useful to include experiments or comparisons to assess whether this sampling step leads to a significant increase in computational cost.

3. Lacking the ablation study about the different backbones. Since the paper aims to provide a theoretical framework for VLMs, it is surprising that the experiments focus solely on CLIP as the backbone. Including comparisons with other VLMs would strengthen the validity of the claims and provide a more comprehensive evaluation of the proposed method

To sum up, this paper leverages the information theory by connecting the PMI and energy function to detect OOD samples, which is interesting. However, due to the weaknesses, the contribution of this paper is limited. Meanwhile, this paper lacks the necessary ablation study. Thus, I rate it as borderline reject. If the author could address these concerns, I am willing to increase my score.

---

> ### Author Rebuttal · Authors · 2025-07-29
>
> We appreciate the insightful comments provided by Reviewer oce3. Please see our responses to your concerns below.
>
> ## 1. The authors have not clearly distinguished their approach from [a]
>
> We respectfully point out key distinctions that demonstrate the novelty of our work:
>
> - Multi-Modal Energy as PMI Estimation
>
> [a] defines energy over unimodal pre-trained classifier logits. In stark contrast, we derive an information-theoretic energy function directly from the Pointwise Mutual Information (PMI) between images and text embeddings in pre-trained VLMs (e.g., CLIP). This provides the first unified framework explaining established CLIP-based scores (MCM, NegLabel) as PMI estimators, establishing a novel theoretical grounding for post-hoc VLM-based OOD detection and directly motivates our method.
>
> It is non-trivial to extend energy-based scoring function from single-modal to multi-modal settings. Clearly, a natural approch to define the energy based on the CLIP-based zero-shot classifier logits, i.e., cosine similarity. However, as thorectically justified in [e], it is PMI not cosine similarity that correspondes to the optimal similarity measure in the CLIP's uni-modal embedding space. This is also empirically supported by empirical observations in [b], i.e., Energy (zero-shot) achieves significantly worse performance (79.57 AUROC and 82.21 FPR95 in average) than MCM and NegLabel, both of which, as proved in the manuscript, can be understood as different PMI estimators.
>
>
> - Divide-and-Conquer Chain Rule Decomposition
>
> Rather than estimating PMI in one shot, we decompose it via the chain rule into two simpler terms，this provably raises the estimation upper bound and therefore reduces underestimation bias while keeping the number of selected negative labels unchanged. We note that This decomposition is unique to our framework and not possible in prior energy methods.
>
> Our contributions are:
>
> (i) A new PMI-theoretic foundation for multi-modal energy
>
> (ii) A novel chain-rule decomposition mitigating PMI estimation bias
>
> This represents a clear conceptual and practical advancement beyond [a] and related works. We thank the reviewer for the opportunity to clarify our novel contributions.
>
> ## 2. Lacking an ablation study about the computation cost.
>
> Thanks for your constructive comments. Since, same as NegLabel [b], this paper focuses on post-hoc OOD detection, our method have no computational cost for training. Regarding the computational cost for inference, the table below reports the inference speed, where the inference speed is measure by Frame Per Second (FPS) with a NVIDIA V100 GPU. It can be found that our method achieves the better trade-off between efficiency and effectiveness.
>
> ||FPS|FPR95|AUROC|
> |-|-|-|-|
> |Ours|558|22.36|96.00|
> |NegLabel|592|25.40|94.21|
> |AdaNeg+NegLabel|476|18.92|96.66|
>
> ##  3. Including comparisons with other VLMs would strengthen the validity of the claims.
>
> Thanks your insightful comments. In Appendix H.2, following NegLabel [b], we evalute our method with GroupViT [c]. For your convenvience, the OOD detection results are reported as follows, where our method consistently outperforms NegLabel beyond CLIP-based models .
>
> |FPR95|iNaturalist|SUN|Places|Textures|AVG|
> |-|-|-|-|-|-|
> |NegLabel|8.60|35.12|41.63|47.06|33.10|
> |Ours|6.86|25.08|37.44|46.15|29.08|
>
> |AUROC|iNaturalist|SUN|Places|Textures|AVG|
> |-|-|-|-|-|-|
> |NegLabel|98.07|91.52|88.85|88.45|91.72|
> |Ours|99.22|95.52|92.24|91.28|94.70|
>
> Besides, we conduct extra experiments using ALIGN [f], following NegLabel [b]. We provide the results in the below, where our method keeps outperforming NegLabel.
>
> |FPR95|iNaturalist|SUN|Places|Textures|AVG|
> |-|-|-|-|-|-|
> |NegLabel|4.55|43.18|61.01|49.89|39.66|
> |Ours|3.24|40.21|57.32|45.79|36.64|
>
> |AUROC|iNaturalist|SUN|Places|Textures|AVG|
> |-|-|-|-|-|-|
> |NegLabel|98.86|90.93|85.03|88.24|90.76|
> |Ours|99.15|92.11|88.17|90.46|92.47|
>
> [a] Energy-based out-of-distribution detection
>
> [b] Negative label guided ood detection with pretrained vision-language models
>
> [c] Groupvit: Semantic segmentation emerges from text supervision
>
> [d] AdaNeg: Adaptive Negative Proxy Guided OODDetection with Vision-Language Models
>
> [e]  Weighted Point Set Embedding for Multimodal Contrastive Learning Toward Optimal Similarity Metric, ICLR25
>
> [f] Scaling Up Visual and Vision-Language Representation Learning With Noisy Text Supervisio

---

> > ### Comment · Reviewer_oce3 · 2025-08-04
> >
> > Thanks to the author`s rebuttal. I have carefully checked all the contents. The author addresses my concerns about the contribution and ablation study about different backbones. One problem remains with the computation cost. Table in Rebuttal 2 shows that AdaNeg+NegLabels achieves the best trade-off since the metrics are SOTA while only needing 476 FPS, which is better than PMI. Can the author report additional experiments about the Ours+AdaNeg setting? This should further clarify that PMI achieves the best trade-off.

---

> ### Author Response · Authors · 2025-08-04
>
> We're grateful for your feedback during this busy period. As requested, we have conducted additional experiments to evaluate the Ours+AdaNeg setting. The results below demonstrate that our method combined with AdaNeg achieves a better trade-off between speed and performance. While we admit that our method makes FPS slightly decrease, the significant performance improvement justifies this marginal cost. Please be free to let us know if you have any questions. We are happy to address.
> | | FPS|FPR95| AUROC |
> | -| -| - | - |
> | NegLabel+AdaNeg|476| 18.92 | 96.66 |
> | NegLabel+Ours|447| 16.52 | 97.51 |

---

> > ### Comment · Reviewer_oce3 · 2025-08-05
> >
> > Thanks to the author`s feedback. The new experiments demonstrate that PMI can improve the performance of OOD detection, albeit with a slight increase in computational cost. Considering this paper connects the information theory and CLIP, I will increase my score to boardline accept.

---

> > > ### Author Response · Authors · 2025-08-05
> > >
> > > Thank you so much for your recognition of our work and your constructive comment, which is very helpful in improving the quality of our work.

---

### Official Review · Reviewer_X4sW · 2025-07-02

**Clarity:** 2
**Significance:** 2
**Originality:** 2
**Rating:** 4
**Confidence:** 3

**Summary:**

This paper provides a theoretical view of post-hoc CLIP-based OOD detection methods from a point-wise mutual information perspective. It demonstrates that two representative methods in this area, MCM and NegLabel, can both be interpreted as Monte Carlo estimators of the energy function, and suggests theoretical motivation of incorporating negative labels. Then, following the framework, the paper proposes to introduce the use of subviews by randomly masking or cropping parts of an image, to enhance OOD detection performance. Empirical results show that the proposed approach brings consistent performance gain over existing baselines.

**Questions:**

* It is not entirely clear how this paper’s proposed theoretical framework is specific to CLIP-based post-hoc OOD detection. While MCM and NegLabel consider the CLIP-based setting due to its generalizability and emergent properties, their core ideas often mirror traditional OOD detection methods. For instance, MCM can be considered as EnergyScore [1] but with CLIP replacing standard image classifiers, and NegLabel can also in some sense be viewed as a variant of contrastive/outlier exposure based approach. It appears that in this paper’s theoretical framework, CLIP is not specifically relevant except for setting it as the critic. Thus, it would be helpful to provide clarifications on the motivation behind focusing specifically on CLIP-based post-hoc OOD detection, instead of considering a more general OOD detection setting.

* The current theoretical formulation, especially for NegLabel, is somewhat coarse-grained, in the sense that it primarily models the incorporation of non-ID prompts, but does not account for their specific properties or "negativeness". In practice, the selection of negative prompts has a significant impact on performance. However, from the theoretical perspective, it seems that incorporating any kind of non-ID prompt would help over MCM. It would be helpful to see more clarification or discussion on this point.

[1] Liu et al. "Energy-based Out-of-distribution Detection." Neurips 2020.

**Ethical Concerns:**

["NO or VERY MINOR ethics concerns only"]

**Final Justification:**

Thank the authors for the rebuttal. I am satisfied with the response and decide to keep my score of borderline accept.

**Limitations:**

Yes the authors have discussed potential limitations.

**Paper Formatting Concerns:**

No major formatting concern.

**Quality:**

3

**Strengths And Weaknesses:**

Strengths:
* The paper provides a theoretical framework for post-hoc CLIP-based OOD detection, providing a systematic way to analyze methods and complementing the often empirical nature of research in this direction.
* The experiments are extensive, covering challenging settings and compatibility with both training free and prompt learning methods. Results show that the proposed method shows strong performance and brings consistent improvement over the baselines.

Weaknesses:
* It is not entirely clear how the proposed theoretical framework is specifically connected to CLIP-based OOD detection besides the fact that CLIP is used as the estimator. It would be helpful to provide clarifications on the motivation behind focusing specifically on CLIP-based post-hoc OOD detection, instead of examining a more general OOD detection setting.  (See Questions section below.)

* Aspects of the theoretical derivation are built upon various existing works cited in the paper, so there is some contribution offset in this sense.

---

> ### Author Rebuttal · Authors · 2025-07-31
>
> We appreciate the insightful comments provided by Reviewer X4sW. Please see our responses to your questions below.
>
> ## 1. It would be helpful to provide clarifications on the motivation behind focusing specifically on CLIP-based post-hoc OOD detection
>
> Indeed, we argue that our thorectical framework is CLIP-specific. In the following, we will show this point by proving that a pretrained CLIP is naturally a PMI estimator.
>
> > We start from the following following inequality [a]
> >
> >$$F=\mathbb{E}\_{(\mathbf{x},y)\sim\mathbb{P}\_{XY}}[\mathcal{W}(\mathbf{x},y)]=\mathbb{E}\_{(\mathbf{x},y)\sim\mathbb{P}\_{XY}}[\log\frac{p_{Y|X}(y|\mathbf{x})}{p_{Y}(y)}]\geq\mathbb{E}\_{(\mathbf{x},y)\sim\mathbb{P}\_{XY}}[\log\frac{\hat{p}\_{Y|X}(y|\mathbf{x};\boldsymbol{\theta}')}{p\_{Y}(y)}]=F_{\boldsymbol{\theta}'} $$
> >
> >Clearly, the inequality holds with inequality if and only if $\hat{p}\_{Y|X}(y|\mathbf{x};\boldsymbol{\theta}')=p_{Y|X}(y|\mathbf{x})$.
>
> > Recalling a CLIP training dataset $\mathcal{D}=${$(\mathbf{x}\_1,y\_1),\ldots,(\mathbf{x}\_N,y\_N)$ } $\sim\mathbb{P}^N\_{XY}$ and CLIP learning objective
> >$$L^{CLIP}\_{\boldsymbol{\theta}'}=\frac{1}{N}\sum\_{i=1}^N\log\frac{\exp h\_{\boldsymbol{\theta}}(\mathbf{x}_i,y_i)}{\sum\_{j=1}^N \exp h\_{\boldsymbol{\theta}}(\mathbf{x}\_j,y\_j)}$$
>
> > For the MCM case, by parameterizing $\hat{p}\_{Y|X}(y|\mathbf{x};\boldsymbol{\theta}')$ via Eq. (7), we have
> >$$F_{\boldsymbol{\theta}'}^{MCM}=\mathbb{E}\_{(\mathbf{x},y)\sim\mathbb{P}\_{XY}}\left [\log\frac{\exp h\_{\boldsymbol{\theta}}(\mathbf{x},y)}{\mathbb{E}\_{\hat{y} \sim \mathbb{P}\_{Y}}\left [\exp h\_{\boldsymbol{\theta}}(\mathbf{x},\hat{y})\right ]} \right]$$
> > One can easily check that $L^{CLIP}\_{\boldsymbol{\theta}'}$ is an empricial version of $F_{\boldsymbol{\theta}'}^{MCM}$ (by the law of large number)
>
> > For the NegLabel case, by parameterizing $\hat{p}\_{Y|X}(y|\mathbf{x};\boldsymbol{\theta}')$ via Eq. (12) and Eq. (13), we have
> >$$F_{\boldsymbol{\theta}'}^{NegLabel }=\mathbb{E}\_{(\mathbf{x},y)\sim\mathbb{P}\_{XY}}\bigg[\mathbb{E}\_{\hat{\mathbf{y}}\_{N-1}\sim\mathbb{P}\_Y^{N-1}}\bigg[\log\frac{\exp{h_{\boldsymbol{\theta}}(\mathbf{x},y)}}{\exp{h_{\boldsymbol{\theta}}(\mathbf{x},y)}+\sum\_{j=1}^{N-1}\exp{ h\_{\boldsymbol{\theta}}(\mathbf{x},\hat{y}\_j)}} \bigg] \bigg]+\log N$$
> >$F\_{\boldsymbol{\theta}'}^{NegLabel }$ is known as InfoNCE [b]. It has been widely acknowledge in the literature that contrastive learning optimizes InfoNCE.
>
> > Based on the discussion, one may conclude that maximizing $L^{CLIP}\_{\boldsymbol{\theta}'}$ is equivalent to maximizing $F_{\boldsymbol{\theta}'}$ no matter whether $\hat{p}\_{Y|X}(y|\mathbf{x};\boldsymbol{\theta}')$ is parameterized via Eq.(7) or Eq. (12).
> >
> > Thanks to the large-scale nature of the CLIP training dataset and the CLIP model itself, it is reasonable to approximate $F$ with $F_{\boldsymbol{\theta}}$ where $\boldsymbol{\theta}=\arg\max_{\boldsymbol{\theta}'}L^{CLIP}\_{\boldsymbol{\theta}'}$.
> >
> > This implies that we can approximate the PMI $\mathcal{W}(\mathbf{x},y)=\log\frac{{p}\_{Y|X}(y|\mathbf{x})}{p\_{Y}(y)}$ with $\log\frac{\hat{p}\_{Y|X}(y|\mathbf{x};\boldsymbol{\theta})}{p\_{Y}(y)}$
>
>
> ## 2. It seems that incorporating any kind of non-ID prompt would help over MCM.
>
> Yes. Empirical evidence confirms that any non-ID prompts (including randomly selected negative labels) significantly enhance OOD detection performance compared to MCM [c]. As demonstrated in Table 7 (Appendix A.4) of NegLabel [d], using 10,000 randomly chosen labels from WordNet yields superior results:
>
> |FPR95|iNaturalist|SUN|Places|Textures|AVG|
> |-|-|-|-|-|-|
> |MCM|32.20|38.80|46.20|58.50|43.93|
> |NegLabel (random selection)|9.11|28.67|45.10|55.87|34.69|
>
> |AUROC|iNaturalist|SUN|Places|Textures|AVG|
> |-|-|-|-|-|-|
> |MCM|94.59|92.25|90.31|86.12|90.82|
> |NegLabel (random selection)|97.96|93.93|89.54|86.62|92.01|
>
> Even though, we never deny that the quality of negative labels has a significant impact on performance, which can be attributed to semantic consistency in representation space, i.e., inputs with similar semantics cluster closer in the embedding space. In other words, Introducing lower-discriminability negative labels (i.e., those semantically similar to ID classes) focuses the sampling distribution on a narrower region, therefore resulting in a more biased Monte-Carlo estimation.
>
>
> ## 3. the theoretical derivation are built upon existing works cited in the paper
>
> While we agree that our theoretical analysis is partially inspired by foundational principles and estimation techniques from prior work, we respectfully argue that our contribution lies in how we extend these ideas to provide novel insights tailored to OOD detection with vision-language models (VLMs), which is not incremental as it solves the lack of theoretical grounding in CLIP-based OOD detection and directly enables our state-of-the-art method.
>
> - To our best knowledge, we are the first to propose a PMI decomposition via sub-views (Theorem 3) for OOD detection. This decomposed formulation is not only theoretically elegant but also leads to provable benefits in bias reduction (Theorem 4).
>
> - We provide the first formal information-theoretic explanation that unifies MCM and NegLabel as stochastic estimators of PMI (Theorems 1 and 2), bridging previously disconnected empirical techniques under a energy-based view (Section 3.1 & 3.2).
>
> - We argue that it is non-trivial to extend energy-based scoring function from single-modal to multi-modal settings. Clearly, a natural approch to define the energy based on the CLIP-based zero-shot classifier logits, i.e., cosine similarity. However, as thorectically justified in [e], it is PMI not cosine similarity that correspondes to the optimal similarity measure in the CLIP's uni-modal embedding space. This is also empirically supported by empirical observations in [d], i.e., Energy (zero-shot) achieves significantly worse performance (79.57 AUROC and 82.21 FPR95 in average) than MCM and NegLabel, both of which, as proved in the manuscript, can be understood as different PMI estimators.
>
> [a] The im algorithm: A variational approach to information maximization
>
> [b] Representation learning with contrastive predictivecoding
>
> [c] Delving into out-of-distribution detection with vision-language representations
>
> [d] Negative label guided ood detection with pretrained vision-language models
>
> [e] Weighted Point Set Embedding for Multimodal Contrastive Learning Toward Optimal Similarity Metric, ICLR25

---

> > ### Comment · Reviewer_X4sW · 2025-08-05
> >
> > Thank the authors for the rebuttal. I am satisfied with the response and decide to keep my score of borderline accept.

---

> > > ### Author Response · Authors · 2025-08-05
> > >
> > > Thank you so much for your recognition of our work and your constructive comment, which is very helpful in improving the quality of our work.

---

### Official Review · Reviewer_p925 · 2025-07-03

**Clarity:** 3
**Significance:** 2
**Originality:** 2
**Rating:** 4
**Confidence:** 4

**Summary:**

This paper provides a theoretical perspective on CLIP-based OOD detection by interpreting it as stochastic PMI estimation, and enhances performance and robustness through a divide-and-conquer approach using the chain rule of PMI.

**Questions:**

- Evaluate the effect of different sub-view generation methods on OOD detection performance to assess how sensitive the chain-rule decomposition is to the type of augmentation used.
- Test the proposed method on OOD benchmarks from domains other than natural images, such as medical, satellite (e.g., EuroSAT), or sketch datasets, to evaluate whether heuristic sub-view generation still holds in these settings.
- Provide quantitative analysis on inference time and GPU memory usage (e.g., FLOPs, runtime per image) compared to baselines like MCM or NegLabel to demonstrate practical viability of the proposed decomposition strategy.
- Include comparisons with recent OOD detectors (HFTT (NeurIPS 2024), NPOS (ICLR 2023), ZOC (AAAI 2022), etc.) to better position the proposed method in the broader OOD detection landscape.

**Ethical Concerns:**

["NO or VERY MINOR ethics concerns only"]

**Final Justification:**

While I still consider the paper to be borderline (in some points), after reviewing the authors’ rebuttal (additional results), I have decided to raise my score.

**Limitations:**

yes

**Paper Formatting Concerns:**

no formatting issues found

**Quality:**

3

**Strengths And Weaknesses:**

Strengths
- The proposed method leverages the chain rule of PMI to decompose the original estimation into simpler conditional and unconditional components. This divide-and-conquer approach not only reduces estimation complexity but also provably increases the upper bound.
- The method achieves state-of-the-art performance across multiple OOD benchmarks, including ImageNet-S, ImageNet-A, and hard OOD tasks like SSB-hard and NINCO. Notably, it shows improvements in both AUROC and FPR95.

Weaknesses
- The decomposition of PMI using the chain rule depends on heuristic-based transformations such as Cutout or Random Crop to generate sub-views. These augmentations, while effective, lack theoretical justification and may not generalize well across different domains or modalities.
- The decomposition strategy introduces additional computation for PMI estimation, yet the paper does not discuss the impact on inference latency or memory overhead, which is particularly important in real-world post-hoc detection settings.
- While the paper includes many CLIP-based and post-hoc baselines (e.g., MCM, NegLabel, AdaNeg), the overall scope of comparison feels limited.

---

> ### Author Rebuttal · Authors · 2025-07-29
>
> We appreciate the insightful comments provided by Reviewer p925. Please see our responses to your concerns below.
>
> ## 1. Evaluate the effect of different sub-view generation methods
>
>  Indeed, we have conducted studies in Appendix H.3 of the manuscript with regards to different sub-view generation methods including cutout and cropping (the our best knowledge, they are the only two options for image data). For your convenvience, we report the detailed results in the below, where it can be found that cutout and cropping achieve comparable OOD detection perfomance while significantly outperforming NegLabel. This implies the flexibility of our method.
>
> |FPR95|iNaturalist|SUN|Places|Textures|AVG|
> |-|-|-|-|-|-|
> |Cutout|1.26|19.08|32.44|39.38|23.04|
> |Cropping|1.04|18.45|31.15|38.79|22.36|
> |NegLabel|1.91|20.53|35.59|43.56|25.40|
>
> |AUROC|iNaturalist|SUN|Places|Textures|AVG|
> |-|-|-|-|-|-|
> |Cutout|99.52|95.08|95.46|91.50|95.39|
> |Cropping|99.64|96.32|95.81|92.15|96.00|
> |NegLabel|99.64|95.49|91.64|90.22|94.21|94.21|
>
> ## 2. Test the proposed method on OOD benchmarks from domains other than natural images
>
> In fact, we have conducted experiments on the ImageNet-sketch dataset in Table 2 where ImageNet-sketch is abbreviated as ImageNet-S.  For your convenvience, we provide the results in the below.
> |FPR95|iNaturalist|SUN|Places|Textures|AVG|
> |-|-|-|-|-|-|
> |Ours|1.62|20.17|34.69|42.94|25.11
> |NegLabel|2.24|22.73|38.62|46.10|27.42
>
> |AUROC|iNaturalist|SUN|Places|Textures|AVG|
> |-|-|-|-|-|-|
> |Ours|99.51|96.02|93.89|91.35|95.52|
> |NegLabel|99.34|94.93|90.78|89.29|93.59|
>
>
> As per your advice, we additionally conduct experiments on the BIMCV-COVID19+ dataset [a], following the same setup as AdaNeg [b]. Specifically, we select BIMCV as the ID dataset, which includes chest X-ray images CXR (CR, DX) of COVID-19 patients and healthy individuals. For the OOD datasets, we, following AdaNeg [b], use CT-SCAN and X-Ray-Bone datasets. The CT-SCAN dataset includes computed tomography (CT) images of COVID-19 patients and healthy individuals, while the X-Ray-Bone dataset contains X-ray images of hands. We report AUROC as follows:
> |AUROC|CT-SCAN|X-Ray-Bone|AVG|
> |-|-|-|-|
> |ours|74.63|99.82|87.26|
> |NegLabel|63.53|99.68|81.61|
> |ours+AdaNeg|95.16|99.99|97.58|
> |NegLabel+AdaNeg|93.48|99.99|96.74|
>
> ## 3. Provide quantitative analysis on inference time
>
> Regarding the computational cost for inference, the table below reports the inference speed, where, same as AdaNeg [b] the inference speed is measure by Frame Per Second (FPS) with a NVIDIA V100 GPU. It can be found that our method achieves the better trade-off between efficiency and effectiveness.
>
> ||FPS|FPR95|FPR95|
> |-|-|-|-|
> |Ours|558|22.36|96.00|
> |NegLabel|592|25.40|94.21|
> |AdaNeg+NegLabel|476|18.92|96.66|
>
> ##  4. Include comparisons with recent OOD detectors
>
> We thank the reviewer for bringing HFTT, NPOS and ZOC into our eyes. Finding that these works are interesting and make a solid contribution to OOD detection, we are greatly glad to compare them.
>
> In particular, while both HFTT, NPOS and ZOC offer creative directions for OOD detection using vision-language models, our work provides a fundamentally deeper contribution by establishing a unified information-theoretical framework that explains and generalizes existing CLIP-based post-hoc methods. Unlike HFTT, which relies on a specific training scheme and synthetic textual data, or ZOC/NPOS, which introduces an additional generation module requiring supervised training, our method is training-free, fully post-hoc, and compatible with any pretrained CLIP-like model. More importantly, we offer provable theoretical insights into why and how exsiting representative CLIP-based OOD scoring function scores can be interpreted as estimators of point-wise mutual information (PMI), and we propose a novel PMI decomposition strategy that improves estimation fidelity without introducing additional negative labels. This level of theoretical rigor and generality not only bridges gaps in the current literature.
>
> The empirical comparsion below shows that our method achives significantly better performance than others, which empirically highlights the superority of our method.
> |FPR95|iNaturalist|SUN|Places|Textures|AVG|
> |-|-|-|-|-|-|
> |ZOC|87.30|81.5|73.06|98.90|85.19|
> |NPOS|16.58|43.77|45.27|46.12|37.93|
> |HFTT|27.44|19.24|43.54|43.08|33.33|
> |Ours|1.04|18.45|31.15|38.79|22.36|
>
> |AUROC|iNaturalist|SUN|Places|Textures|AVG|
> |-|-|-|-|-|-|
> |ZOC|87.30|81.51|73.06|98.90|85.19|
> |NPOS|96.19|90.44|89.44|88.80|91.22|
> |HFTT|93.27|95.28|90.26|88.23|91.76|
> |Ours|99.64|96.32|95.81|92.15|96.00|
>
> [a] Bimcv covid-19+: a large annotated dataset of rx and ct images from covid-19 patients
>
> [b] AdaNeg: Adaptive Negative Proxy Guided OODDetection with Vision-Language Models

---

> ### Author Response · Authors · 2025-08-05
>
> Dear Reviewer p925:
>
> We're grateful for your mandatory acknowledgement during this busy period. As the rebuttal period is ending soon, please be free to let us know whether your concerns have been addressed or not, and if there are any further questions.
>
> Thanks,
>
> Authors.

---

> > ### Comment · Reviewer_p925 · 2025-08-08
> >
> > Thank you for the detailed rebuttal and the additional clarifications provided. The authors have addressed most of my initial concerns with experimental evidence. After reviewing the authors’ rebuttal and taking into account the comments from other reviewers, I have decided to raise my score.

---

> > > ### Author Response · Authors · 2025-08-08
> > >
> > > Thank you so much for your recognition of our work and your constructive comment, which is very helpful in improving the quality of our work.

---

### Decision · Program_Chairs · 2025-09-17

**Decision:**

Accept (poster)

**Comment:**

Reviewers agreed on several strengths of the paper, including its solid theoretical grounding (p925, X4sW, oce3) and strong empirical performance across benchmarks (p925, X4sW, jjLB). At the same time, concerns were raised in the initial reviews: computational cost (p925, oce3), limited breadth of experiments and analysis (p925, oce3, jjLB), and lack of clarity regarding the theoretical novelty (X4sW, oce3, jjLB). However, the authors' rebuttal and subsequent author-reviewer discussion were generally effective in addressing these points. In their final justifications, all reviewers indicated that their major concerns had been resolved.

Taking all reviews and discussions into account, while the theory in this work explicitly covers only energy-based post-hoc methods, energy-based scoring remains one of the dominant approaches in OOD detection. On this basis, I find that the paper makes a meaningful and timely contribution by offering a unifying theoretical view of energy-based CLIP methods and demonstrating practical performance gains.